# Transition metal dichalcogenide metaphotonic and self-coupled polaritonic platform grown by chemical vapor deposition

Fuhuan Shen[1,5], Zhenghe Zhang[2,3,5], Yaoqiang Zhou[1,5], Jingwen Ma[1], Kun Chen[4], Huanjun Chen [4], Shaojun Wang [2,3] ✉, Jianbin Xu [1] ✉ & Zefeng Chen [1,2,3] ✉

Transition metal dichalcogenides (TMDCs) have recently attracted growing attention in the fields of dielectric nanophotonics because of their high refractive index and excitonic resonances. Despite the recent realizations of Mie resonances by patterning exfoliated TMDC flakes, it is still challenging to achieve large-scale TMDC-based photonic structures with a controllable thickness. Here, we report a bulk $MoS_2$ metaphotonic platform realized by a chemical vapor deposition (CVD) bottom-up method, supporting both pronounced dielectric optical modes and self-coupled polaritons. Magnetic surface lattice resonances (M-SLRs) and their energy-momentum dispersions are demonstrated in 1D $MoS_2$ gratings. Anticrossing behaviors with Rabi splitting up to 170 meV are observed when the M-SLRs are hybridized with the excitons in multilayer $MoS_2$. In addition, distinct Mie modes and anapole-exciton polaritons are also experimentally demonstrated in 2D $MoS_2$ disk arrays. We believe that the CVD bottom-up method would open up many possibilities to achieve large-scale TMDC-based photonic devices and enrich the toolbox of engineering exciton-photon interactions in TMDCs.

The rapid progress in nanophotonics has a profound influence on an abundance of fields such as nonlinear optics and quantum photonics[1–6], forming the bases of multiple photonic metadevices, including single-photon switches, nano-scale lasers, and metasurfaces[3,7–9]. Compared to the plasmonic counterparts which suffer from high metal losses and heating problems, the dielectric nanostructures offer unique possibilities to confine resonant modes in a subwavelength scale with reduced material losses[10–12]. Importantly, the coexistence of magnetic and electric resonances of the dielectric resonators, and their interference, bring intriguing functionalities to the system such as unidirectional scattering in the far field and enhanced nonlinear responses[13–15]. Moreover, by artificially designing and arranging the optical components, the wavefront shaping of light could be achieved, leading to promising applications such as metalens and Huygens metasurfaces[16,17]. Recently, with the topological effects, dielectric metastructures are playing a vital role in the development of novel topological photonics[18,19]. Multiple resonant modes, subwavelength thickness,

[1]Department of Electronic Engineering, The Chinese University of Hong Kong, Shatin, N.T., Hong Kong SAR, P. R. China. [2]School of Optoelectronic Science and Engineering and Collaborative Innovation Center of Suzhou Nano Science and Technology, Soochow University, Suzhou 215006, P. R. China. [3]Key Lab of Advanced Optical Manufacturing Technologies of Jiangsu Province & Key Lab of Modern Optical Technologies of Education Ministry of China, Soochow University, Suzhou 215006, P. R. China. [4]State Key Lab of Optoelectronic Materials and Technologies, Guangdong Province Key Laboratory of Display Material and Technology, School of Electronics and Information Technology, Sun Yat-sen University, Guangzhou 510275, P. R. China. [5]These authors contributed equally: Fuhuan Shen, Zhenghe Zhang, Yaoqiang Zhou. ✉e-mail: swang.opto@suda.edu.cn; jbxu@ee.cuhk.edu.hk; zfchen@ee.cuhk.edu.hk

and design flexibility make the dielectric metastructures a superb platform to manipulate the light field and facilitate the realization of strong light-matter in a compact way[20-22].

On the other hand, semiconducting transition metal dichalcogenides (TMDCs) have emerged as promising van der Waals (vdW) materials because of their high refractive index and distinct excitonic properties at room temperature[12,23-25]. In particular, monolayer (ML) TMDCs such as $MoS_2$ or $WS_2$ are of direct bandgap and exhibit large excitonic oscillator strengths. By coupling ML TMDCs with optical cavities or dielectric photonic crystals (PCs), the half-light half-matter quasiparticles, i.e., exciton-polaritons, will be formed due to the strong exciton-photon interactions[26-28]. However, the atomic thin nature of ML TMDCs (thickness < 1 nm) is nonideal to construct the dielectric resonators by themselves to support optical resonant modes at visible or near-infrared frequencies, due to the limited thickness along the $z$-direction[29,30]. Compared to monolayer counterparts, albeit lacking strong luminescence due to the indirect bandgap, multilayer TMDCs still exhibit relatively large exciton oscillator strengths and high refractive indices (>4)[30]. Distinct Mie modes and self-coupled exciton-polaritons have been reported with a proof-of-concept bulk $WS_2$ nanodisk[11]. Moreover, in recent years, PC modes and enhanced nonlinear emissions have been experimentally demonstrated in the metasurfaces built by high refractive-index TMDCs[31-34], which even show better performances in some aspects when compared with metastructures based on Si or GaAs[34]. The high index, as well as the presence of excitons, renders multilayer TMDCs promising to enrich the existing dielectric materials for nanophotonics and polaritonics[35,36].

To date, the overwhelming majority of previous reports on TMDC nanostructures are based on the multilayer TMDC flakes by a mechanical exfoliation method[11,24,37]. The exfoliated TMDC flakes are of random sizes and suffer from uncontrollable and uneven thickness[38], thus impeding the sufficient reproducibility and large-scale production of the TMDC metastructures. In contrast, the chemical vapor deposition (CVD) techniques have been proven to be efficient fabrication methods to synthesize large-area TMDCs with controllable thickness[37,39]. It is thus of great promise to adopt the CVD bottom-up method to build scalable photonic devices with TMDC materials, to overcome the limitations due to the exfoliated flakes.

Here, we report the experimental realization of a large-area (up to $0.38 \times 0.38$ $mm^2$ in scale) $MoS_2$ metastructure platform via a CVD bottom-up fabrication method. The material properties of bulk $MoS_2$ structures are firstly characterized and the polycrystalline nature of $MoS_2$ (after sulfidation) is shown. We find that the geometric shape and the uniformity of the designed structures are well retained after the sulfidation process. The dielectric resonant modes in both 1D $MoS_2$ grating and 2D $MoS_2$ disk array are demonstrated through the optical measurements, which can be well reproduced with the numerical simulations. In addition, the self-coupled exciton-polaritons with unambiguous anticrossing behaviors are also experimentally observed. We argue that this CVD bottom-up method for dielectric metastructures would enrich the toolbox of engineering exciton-photon interactions in TMDCs.

## Results

### Fabrication of $MoS_2$ metastructures

Figure 1a outlines the procedures to fabricate the $MoS_2$ metastructures (disk array is used as a representative example). Firstly, using standard electron-beam lithography (EBL) and electron-beam evaporation (EBE) technologies, we fabricate the Mo nanopatterns on $SiO_2$ (2 μm)/Si substrate (I-II). Then the patterned Mo structures are sent to the quartz-tube furnace for the sulfidation process at the temperature of 750 °C (III). Finally, the Mo patterns would be chemically converted to the $MoS_2$ patterns after the sulfidation process (IV).

### Structures and material properties characterizations

Figure 1b shows the microscope image of the 1D $MoS_2$ gratings, along with the zoomed-in scanning electron microscopy (SEM) image. The color of the corresponding structure evolves with different periods $P$ and filling factors $\Lambda = \frac{w}{P}$ where $w$ is the width of the original Mo grating bar. Different capital letters (A, B, C...) and numbers (1, 2, 3...) are adopted to label various $P$ and $\Lambda$ respectively. For instance, A1 represents $P = 400$ nm and $\Lambda = 0.3$ and A2 represents $P = 400$ nm and $\Lambda = 0.4$ (see Supplementary Fig. 1 for comprehensive definitions). Supplementary Fig. 5 shows that the geometric shape and uniformity of the grating bars (or disks) are well retained after sulfidation, while slightly lateral extension can be found. The atomic force microscope (AFM) images (Fig. 1c) of Mo pattern and corresponding $MoS_2$ pattern confirm the unaltered period and the lateral extension, but also show the growing-up of the thickness after the sulfidation process. As the height profiles in Fig. 1d show, the height of the original Mo grating bar is increased from $55 \pm 5$ nm (green curve) to $110 \pm 10$ nm (red curve) when converted to $MoS_2$ pattern. Figure 1e–g shows similar results for 2D $MoS_2$ disk arrays whose total device size is up to 0.38 mm (Fig. 1e). Compared to the previous work where the exfoliated $WS_2$ or $MoS_2$ flakes were adopted[11,12,30], our bottom-up method achieves scalable TMDC metastructures with a controlled thickness[40]. In addition, all-dielectric TMDC structures (thickness > 100 nm) show distinct differences from the previously reported Au@$MoS_2$ structure where only a few-layer (2–4 layers) $MoS_2$ is formed at the surface of Au disk[39]. Strong electric and magnetic resonances can thus be realized and enhanced in these well-structured, high refractive-index $MoS_2$ metastructures, which are key to the fields of metaphotonics[41].

Before studying the optical properties of the $MoS_2$ metastructures, we first characterize the structural and material properties of our $MoS_2$ sample after the sulfidation process. Scanning transmission electron microscopy (STEM) and energy dispersive X-ray spectroscopy (EDS) techniques are used to characterize the chemical compositions and crystal structure of the $MoS_2$ samples. Figure 2a shows the STEM images of the cross-section of 1D $MoS_2$ grating (A2), to exhibit the distributions of the elements with the EDS maps. We find that Mo and S elements are uniformly distributed both at the out shell and inside of the grating bar, confirming that the Mo structures are sulfurized to the $MoS_2$ structures completely. High-resolution TEM (HRTEM) images in Fig. 2b exhibit the polycrystalline nature of our $MoS_2$ samples. The interval of neighboring $MoS_2$ layers is around 0.62 nm which is consistent with previous results[42]. More interestingly we find that the $MoS_2$ of around 10 layers prefer to grow horizontally at the surface and the interface between the $MoS_2$ and substrate (green and magenta boxes) while the internal $MoS_2$ layers prefer to be vertically aligned to the surface (yellow box).

$MoS_2$ patterns are then transferred to the transparent glass substrate (Fig. 2c) via a wet transfer method[43] for optical characterizations. The Raman spectra (Fig. 2d) are measured from the top and bottom sides separately, which both show the signatures of two peaks for $MoS_2$ (in sharp comparison with the Raman spectrum of Mo pattern)[44]. Figure 2e shows the refractive index ($n$) and extinction coefficient ($\kappa$) of the bulk $MoS_2$, which can be roughly divided into two regions: region I (400–750 nm) with the unambiguous excitonic resonances, region II (>750 nm) with high refractive index ($n > 4$) and low material loss ($\kappa < 10^{-2}$).

### M-SLR modes in $MoS_2$ gratings

The grating structure is anisotropic in-plane and its optical response is dependent both on the polarization and propagation direction of the incident light, as illustrated in Fig. 3a. $x$ ($y$)-direction is defined as along (across) the bar and the $z$-direction is defined as perpendicular to the grating plane. For the polarization, transverse-magnetic (TM for short)/transverse-electric (TE for short) represents the electric field across/along the bar. Without loss of generality, A1 to A3 are selected

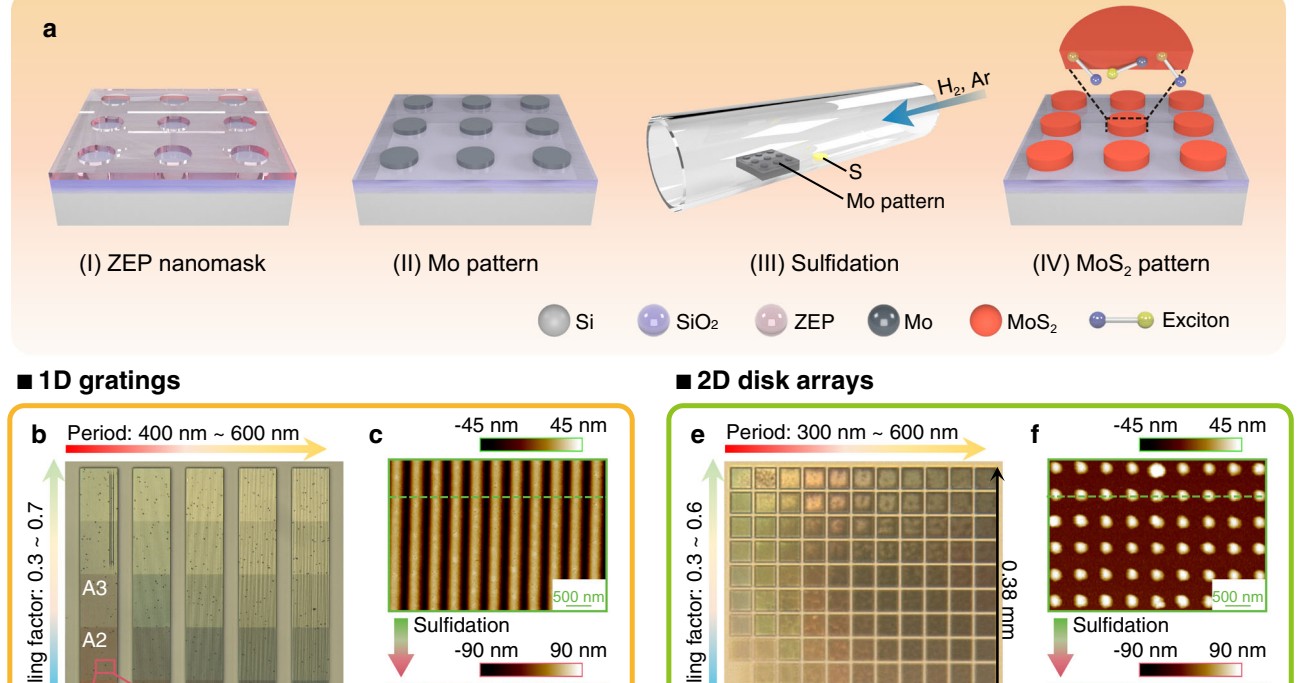

**Fig. 1 | MoS₂ metastructures. a** A schematic diagram to outline the chemical vapor deposition (CVD) bottom-up fabrication process. The photoresist ZEP-520A is patterned by the electron beam lithography, forming the designed ZEP nanomask on the substrate (I). **b** The optical microscope image of 1D MoS₂ gratings with various periods $P$ (from 400 nm to 600 nm with a step of 50 nm) and filling factors $\Lambda$ (from 0.3 to 0.7 with a step of 0.1). A zoomed-in scanning electron microscopy (SEM) image for A2 region is shown at the bottom. **c** Atomic force microscopy (AFM) images of Mo pattern and MoS₂ pattern (A2 region). **d** Height profiles contrast of Mo pattern (green line) and MoS₂ pattern (red line) indicated by the dashed lines in (**c**). **e–g** Same as (**b–d**) but for the 2D MoS₂ disk arrays.

as representative examples (see Supplementary Fig. 9 for extinction spectra of other gratings). From A1 to A3 the period is kept at 400 nm while the width is increased, as clearly shown by the SEM images in Fig. 3b. Figure 3c shows the measured extinction spectra for A1-A3 under normal incidence (TM-polarized) which can be well reproduced by the numerical simulation (see methods). In addition to the A, B excitonic resonances that emerged in the spectra, a surface lattice resonance (SLR) mode, arising from the coupling of the individual MoS₂ bar and the in-plane diffraction orders of the grating lattice, appears on the low-energy side of A exciton[27]. The calculated near-field distributions (insets in Fig. 3c) indicate the magnetic nature of the SLR[45]. This magnetic resonance arises from the phase retardation inside of the particle along the propagation direction (i.e., $z$-direction) such that a displacement current is formed[45]. We label this resonance as magnetic SLR (or M-SLR for short). The comparisons of near-field distributions under TM and TE polaritons are shown in Supplementary Fig. 7.

The linewidth of M-SLR is estimated as $\gamma \approx 100$ meV by Lorentzian fitting[46] (Fig. 3c). The origin of the measured linewidth is basically from two parts: the intrinsic material loss and the radiation loss[47], i.e., $\gamma = \kappa + \gamma_{rad}$. To estimate the material loss ($\kappa$) in M-SLR, we calculate the extinction spectra (Fig. 3d) for a grating structure (inset in Fig. 3d) by adopting various material losses $\kappa$ (from 0 to

0.2) for the grating bar. The period is set as 400 nm and the corresponding refractive index of grating bar is chosen as $n = 4.7$, to match the parameters of the MoS₂ grating structure (A1). Corresponding linewidths are then extracted by fittings (dashed lines in Fig. 3d). As Fig. 3e shows, the extracted linewidth $\gamma$ is linearly related to $\kappa$ while ($\gamma - \kappa$) is almost unchanged (i.e., $\gamma_{rad}$ is constant for the same geometric parameters and refractive index adopted in the grating structure). At zero material loss ($\kappa = 0$), the linewidth is around 96 meV which is purely ascribed to the radiation loss (i.e., $\gamma_{rad} \approx 96 meV$). By inserting linewidth from our experiment result (yellow star), we find that the material loss of M-SLR mode in Fig. 3c is estimated as $\kappa < 10^{-2}$, confirming the high refractive index and low material loss properties of bulk MoS₂ in region II.

**Polariton dispersions in MoS₂ gratings**

The energy-momentum dispersions of MoS₂ gratings are measured by angle-resolved transmission spectra (see methods). Without loss of generality, we here focus on the dispersions of TM-polarized modes along $\mathbf{k}_y$. Results for TE-polarized modes and propagation direction along $\mathbf{k}_x$ are shown in Supplementary Fig. 6. Figure 4a shows the measured dispersions which can be well reproduced with the numerical simulations (Fig. 4b). The theoretical model is applied to describe the dispersions due to the coupling of excitonic resonances of MoS₂

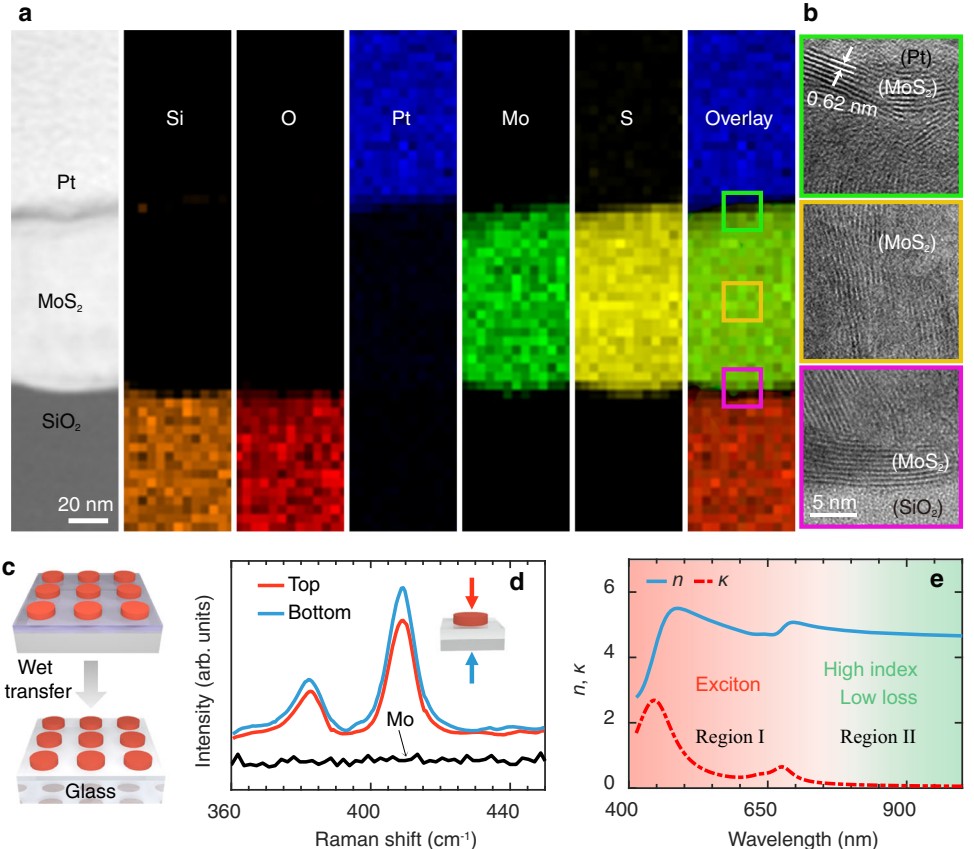

**Fig. 2 | Structural and material characterizations of MoS₂ pattern. a** Cross-sectional low magnification view (left 1) of scanning transmission electron microscopy (STEM) micrograph of the MoS₂ grating for elemental mapping. Corresponding energy dispersive X-ray spectroscopy (EDS) maps of the MoS₂ cross-section show the spatial distributions of elements Si, O, Pt, Mo, S, and overlay separately. **b** High-resolution transmission electron microscopy (HRTEM) graphs of the top boundary (green box), internal part (yellow box), and bottom boundary (magenta box) of the MoS₂ bar. **c** Schematic of the transferring procedure of MoS₂ metastructure from SiO₂ (2 μm)/Si substrate to glass substrate. **d** Raman spectra measured from the top (red curve) and the bottom (blue curve) sides of the MoS₂ structures. The black curve represents the Raman spectrum from Mo patterns. **e** The real (blue solid curve) and imaginary (red dashed-dotted curve) parts of the dielectric function of multilayer MoS₂ after sulfidation. Red and green areas are used to differentiate region I (400 nm to 750 nm) and region II (>750 nm) of the complex refractive index (i.e., $n + i\kappa$) of bulk MoS₂.

and M-SLR modes (Supplementary Note 1, 2):

$$\omega_{\pm} = \frac{\omega_{\mathrm{cav}} + \omega_{\mathrm{A}}}{2} \pm \sqrt{g_{\mathrm{A}}^2 + \frac{\delta^2}{4}}, \tag{1}$$

where $\omega_{\mathrm{cav}}$ and $\omega_{\mathrm{A}}$ represent the resonant frequencies of cavity mode (M-SLR) and A exciton. Detuning of cavity mode and A excitonic resonance is defined as $\delta = \omega_{\mathrm{cav}} - \omega_{\mathrm{exc}}$. $g_{\mathrm{A}}$ represents the coupling strength of A exciton to the cavity mode. Here, similar to the previous work[42,48,49], only A exciton is considered in the coupling for simplicity. As Fig. 4a, b shows, the dispersions of LP (short for lower polariton) and UP (short for upper polariton) can be well described by the theoretical model (+ and − signs are used to separate the polaritons regarding whether SLR (+1) or SLR (−1) mode is coupled to the A exciton). A clear anti-crossing behavior (Fig. 4c) is shown in the zoomed-in regions (A2 and A3) from Fig. 4a.

To give an unambiguous illustration of polariton dispersions observed in Fig. 4a, b, a 1D grating structure (geometric parameters: height = 100 nm, width = 150 nm, period = 400 nm) is adopted for the numerical simulations where the dielectric function of the constituent material is described by a multi-Lorentz oscillator model[11]:

$$\varepsilon = \varepsilon_0 + f \frac{\omega_0^2}{\omega_0^2 - \omega^2 - i\gamma\omega}. \tag{2}$$

$\varepsilon_0 = 16$ is adopted as the background permittivity. $\omega_0 = 2.1$ eV and $\gamma = 40 meV$ are adopted as the resonant frequency and linewidth of the exciton. $f$ is the corresponding excitonic oscillator strength. We adopt three sets of oscillator strengths: (I) $f = 0$ (i.e., without excitonic resonance), (II) $f = 0.2$, and (III) $f = 0.4$, whose corresponding dielectric functions are exhibited in Fig. 4d. Figure 4e shows the calculated angular-resolved extinction spectra for the 1D grating structure with the dielectric functions from Fig. 4d. When $f = 0$, the M-SLR shows the linear dispersion, associated with the ±1 diffraction orders, i.e., Rayleigh anomalies (RAs), which is detailly analyzed in Supplementary Note 2. Such linear dispersions of RAs are also observed in our fabricated samples (Supplementary Fig. 3a) where the M-SLR is far detuned from the excitonic resonances.

However, once the excitons are coupled in the grating structure (i.e., $f > 0$), the originally linear-dispersed M-SLR would strongly interact with the excitons, such that two polaritons states (LP and UP) are generated with a clear anticrossing behavior. By fitting results with the polariton dispersions (Eq.(1)), the Rabi splitting can be calculated as 170 meV (II) and 240 meV (III) which meets the previously reported $\Omega \propto \sqrt{f}$ relation[46]. The above analyses confirm the self-coupled mechanism shown in Fig. 4a, b where the cavity modes (M-SLRs) strongly interact with the excitons inside of the grating structures, resulting in robust and highly dispersive polaritons.

Meanwhile, we surprisingly find that, as Fig. 4f exhibits, the coupling strength $g_{\mathrm{A}}$ is increased from 40 meV (A1), 55 meV (A2) to 85 meV

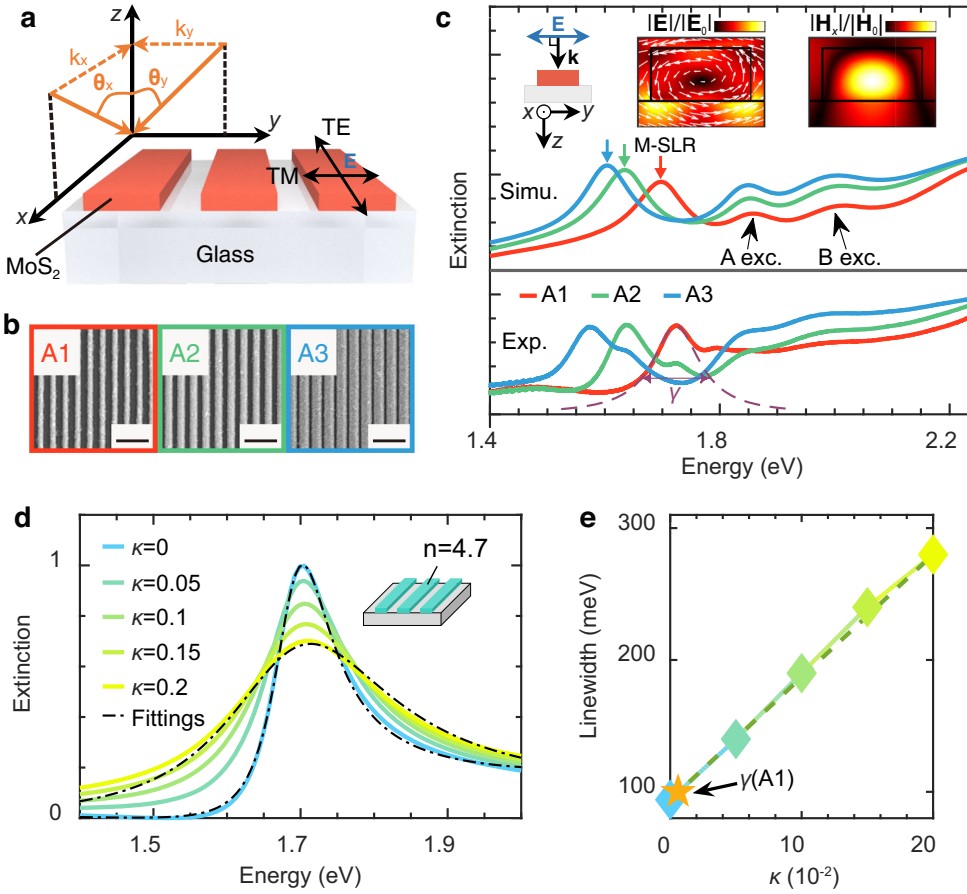

**Fig. 3 | M-SLR mode of the 1D MoS$_2$ grating. a** Schematic structure of MoS$_2$ grating on the glass substrate. $\mathbf{k}_x$ and $\mathbf{k}_y$ are in-plane components of the incident wavevector $\mathbf{k}$ in $x$- and $y$- directions. TM is short for transverse-magnetic polarization and TE is short for transverse-electric polarization. **b** SEM images for 1D MoS$_2$ gratings (A1-A3 regions). All black scale bars represent 1 μm. **c** Measured (bottom) and simulated (top) extinction spectra under TM-polarized incidence for A1-A3 regions. Red, green, and blue arrows indicate the magnetic-surface lattice resonance (M-SLR) modes of A1-A3, where the spatial distributions of the electric and magnetic ($x$-component) fields at M-SLR are shown in the insets. The white arrow represents the real part of vectorial electric field projected in the $yz$-plane. **d** Simulated extinction spectra for 1D dielectric gratings embedded in a homogeneous environment ($n$= 1.46). Different extinction coefficients $\kappa$ (from 0 to 0.2) of dielectric material are adopted in the calculation. Black dashed-dotted lines are fitting curves based on the formula in ref. 55 (here only $\kappa = 0$ and $\kappa = 0.2$ are shown for clarity). In the simulation, the height and width of the grating bar are chosen as 100 nm and 110 nm separately to match the geometric parameters of A1. **e** Extracted linewidth $\gamma$ as a function of the material loss $\kappa$ from the fitting results in (**c**). The yellow star indicates the linewidth of the M-SLR mode from A1 region.

(A3). The previous studies on TMDC-cavity hybrid systems indicated that both the cavity mode volume and the exciton numbers determine systematic Rabi splitting[25–27,49,50]. In our system, the resonator itself is composed of the bulk MoS$_2$ showing the excitonic resonances. Though the increase of the mode volume would degrade the interaction strength between a single exciton and optical modes, the benefit comes with more excitons entering the coupling process. A more quantitative analysis will be shown in the following.

As indicated by Supplementary eqs (10, 11), Rabi splitting $\Omega \propto \frac{\sqrt{N}}{\sqrt{V}}$. Both the number of excitons ($N$) and mode volume ($V$) of the resonant mode determine the coupling strength of the system[1]. In this context, we define the mode area as $\sigma_m = \iint \frac{\varepsilon E^2 ds}{\max[\varepsilon E^2]}$ and the cross-sectional area as $\sigma_c = w_{gra} \times h_{gra}$ ($w_{gra}$ and $h_{gra}$ represent the width and height of the MoS$_2$ grating bar, respectively). One thus can calculate the mode volume $V = \sigma_m \times x$ and exciton number $N \propto (\sigma_c \times x)$ where $x$ is the length along the bar. Compared to the width, the length of the grating bar can be treated as infinite. As a result, the Rabi splitting can be deduced as $\Omega \propto \frac{\sqrt{\sigma_c}}{\sqrt{\sigma_m}}$ (defined as the effective area $\sigma_{eff} = \frac{\sqrt{\sigma_c}}{\sqrt{\sigma_m}}$). Figure 4g shows the calculated $\sigma_c$ and $\sigma_m$ from simulation results of A1 to A3. The increase of $\sigma_m$ reflects the increase of the mode volume of M-SLR, which would reduce the coupling intensity. On the other hand, the increase of $\sigma_c$ indicates more excitons entering the coupling process. The combined effect is reflected by the effective area $\sigma_{eff}$. In Fig. 4h, we

find that $\sigma_{eff}$ shows great consistency with the Rabi splitting (represented by $2g_A$) from A1 to A3, confirming the above analyses.

## Mie modes and polaritons in MoS$_2$ disk arrays

To introduce more possible optical resonant modes, 2D MoS$_2$ square disk arrays are designed and fabricated (Fig. 1e). We choose H1-H7 (period $P$ = 510 nm and filling factor $\Lambda$ varies from 0.3 to 0.48) as representative examples, whose microscope images are shown in Fig. 5a. The measured extinction spectra under normal incidence are shown in Fig. 5b and can be well reproduced by numerical simulations (Fig. 5c). Multiple optical Mie modes appear and their resonant frequencies show a red shift with the increase of the filling factor. The simulated spectrum of H3 is selected for the detailed analyses. As Fig. 5d shows, two resonant peaks overlap with each other in region II (1.25 ~ 1.7 eV). The electric near field distributions in Fig. 5e reflect the electric dipole (ED) and magnetic dipole (MD) instants for these two modes respectively[45]. In region I (>1.7 eV), a broad and asymmetric peak arises and near field distribution in Fig. 5e indicates the anapole-like signature of this mode. We find that the electric field distribution of the anapole mode supported in the square disk array is slightly different from that in the isolated particle[13]. This is due to the mutual interactions of neighboring disks in a 2D array, as is detailed in this work[51].

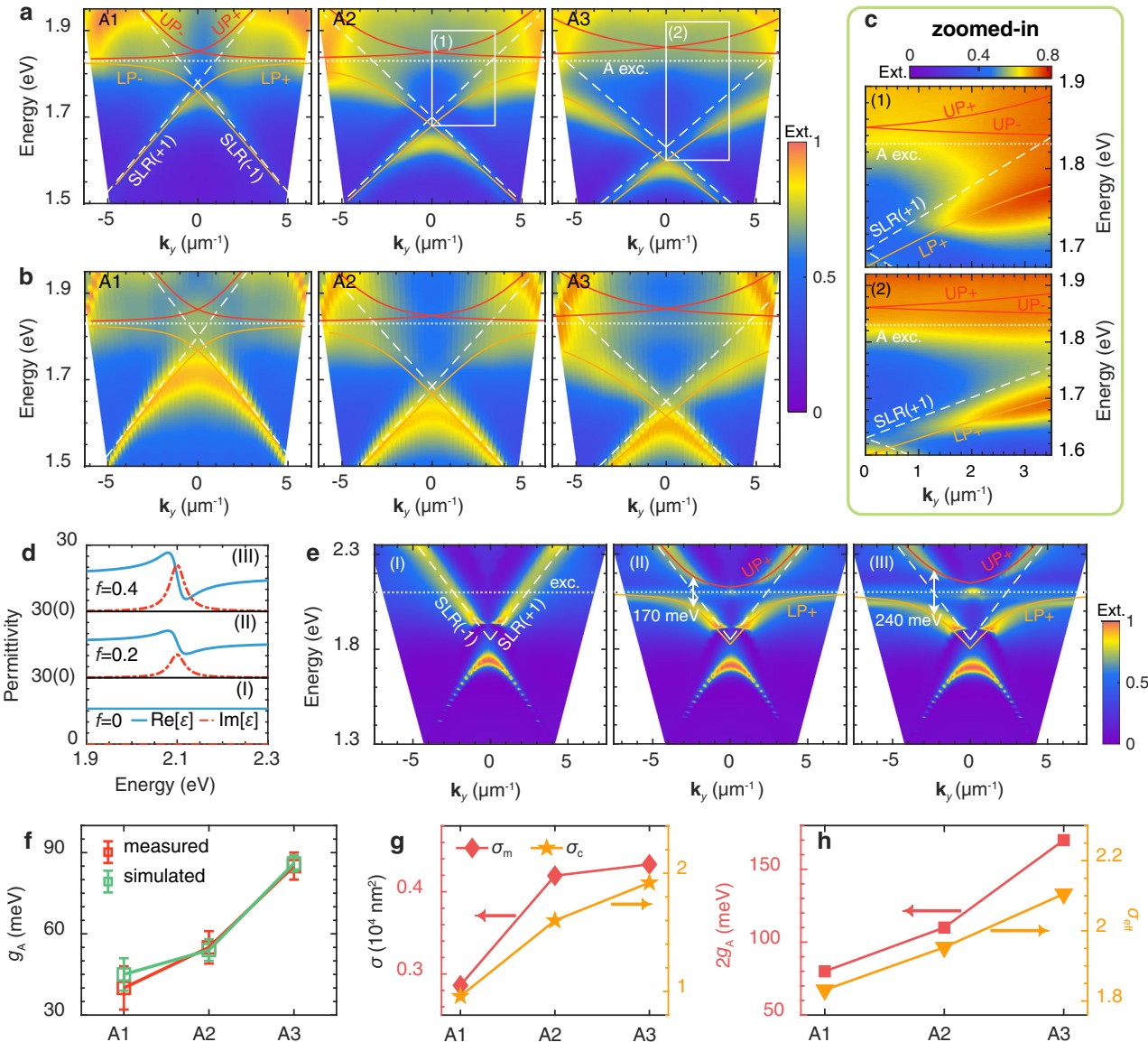

**Fig. 4 | Self-coupled polariton dispersions in 1D MoS₂ gratings. a** Measured and **(b)** simulated angle-resolved extinction spectra for 1D MoS₂ gratings (A1-A3) along $\mathbf{k}_y(\mathbf{k}_x=0)$. The white dashed lines represent the dispersions of surface lattice resonances (SLRs) of the MoS₂ grating where the symbol ±1 represents the ±1 diffraction orders, i.e., the so-called Rayleigh anomalies (RAs) (see Supplementary Note 2 for more details). The white dotted curves indicate the A excitonic resonance. Red and yellow lines represent the dispersions of the upper polariton (UP) and lower polariton (LP) separately, where the +/− symbol represents the polariton formed from the SLR(+1)/SLR(−1) mode. **c** Zoomed-in regions of white box (1) and box (2) indicated in (**a**). **d** Permittivity calculated by the Lorentz model for oscillator strengths $f=0$ (I), $f=0.2$ (II), and $f=0.4$ (III) respectively. **e** Calculated angled-resolved extinction spectra of 1D dielectric grating for different dielectric functions shown in (**c**). **f** Coupling strengths $g_A$ extracted from the polariton dispersions from (**a, b**). **g** Mode areas $\sigma_m$ and cross-sectional areas $\sigma_c$ calculated from the simulation results from (**b**). **h** Comparisons of coupling strengths ($2g_A$ represents the Rabi splitting) and effective areas $\sigma_{eff}$ for A1-A3.

Figure 5f–i shows the zoomed-in extinction spectra (indicated by the box I,II in Fig. 5b, c) for the experimental (Fig. 5f, g) and simulated (Fig. 5h, i) results. With the filling factor (i.e., the diameter of the disk) increasing, the anapole gradually overlaps with the A excitonic resonance, forming two branches (UP, LP) in the extinction spectra. The dispersions of polaritons could be well fitted by the theoretical model (Eq.(1)). The typical Fano interference signature indicates the coupling is within the weak or intermediate coupling regime[52].

## Discussion

To sum up, we present a CVD bottom-up method to fabricate the large-area metastructures based on the high refractive-index MoS₂. In such a TMDC-based metaphotonic platform, both the dielectric resonant modes such as M-SLR and their interactions with excitons are studied

with both experimental measurements and numerical simulations. Moreover, the resonant modes and self-coupled polaritons due to light-matter interaction can be tuned by changing the incidence conditions (e.g., incident angles) and geometric parameters of TMDC structures.

In this work, both the 1D and 2D metastructures are built with the non-luminous material, i.e., the multilayer MoS₂, which impedes further applications such as using it as a gain medium in the field of nanolasers. Extending our CVD bottom-up method to some direct bandgap TMDC materials such as InSe is worthy to be explored in the future[37]. Nevertheless, despite the limitation due to the indirect bandgap of bulk MoS₂, the pronounced dielectric resonances and self-coupled polaritons demonstrated in this work indicate the great potential of adopting the CVD bottom-up method for dielectric metaphotonics and engineering light-matter interaction. The bottom-

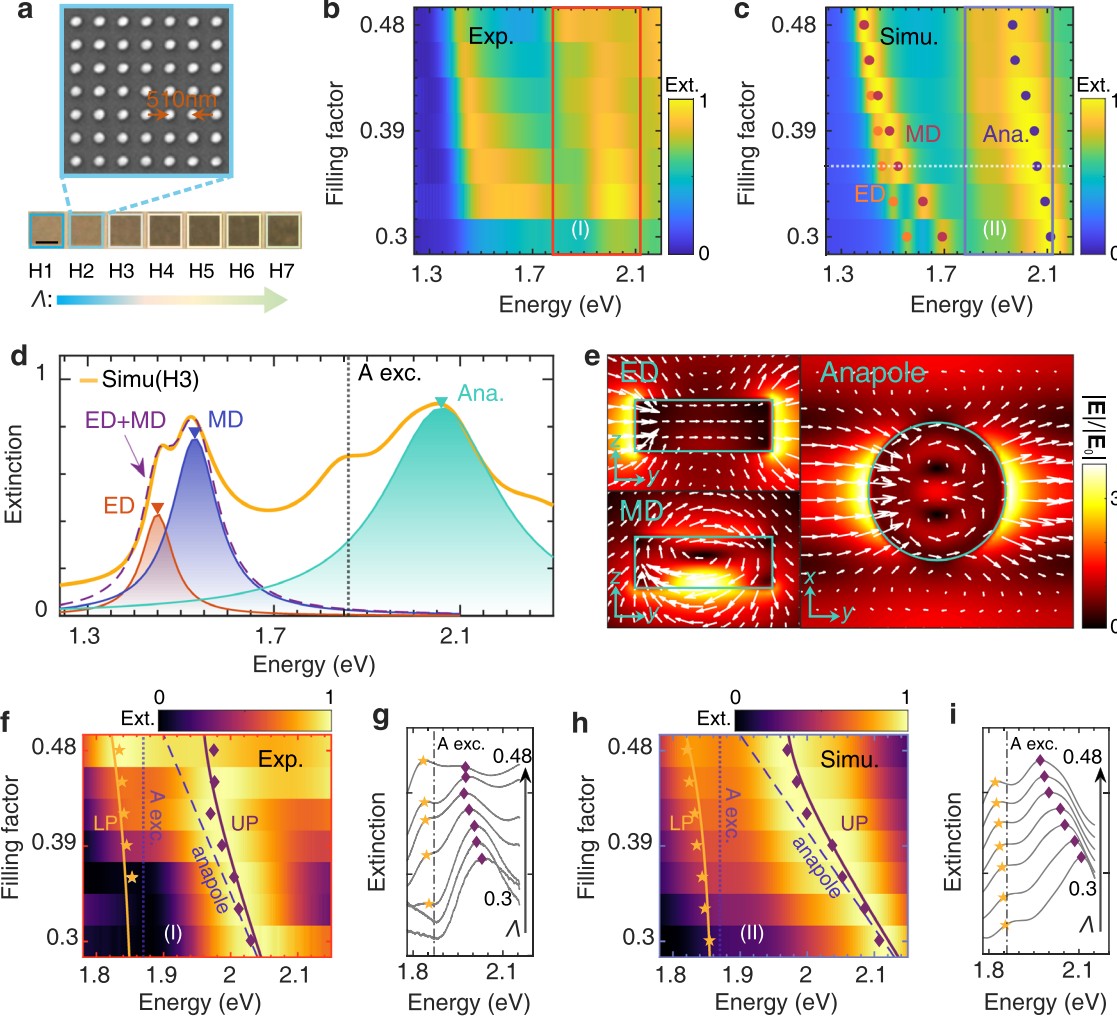

**Fig. 5 | Mie resonant modes and exciton-anapole polaritons in 2D MoS₂ disk arrays. a** Optical microscope images of 2D MoS₂ disk arrays (H1-H7, i.e., $P$=510 nm and $\Lambda$ is from 0.3 to 0.48), along with the SEM image for H2 sample. The scale bar represents 20 μm. **b, c** Measured (**b**) and simulated (**c**) extinction spectra (under normal incidence) for H1-H7. Red, blue, and cyan circles represent the resonant frequencies of electric dipole (ED), magnetic dipole (MD), and anapole modes. **d** The simulated spectrum for H3 region (indicated by the white dotted line in (**c**)). Red, blue, and cyan lines are Lorentzian fittings for ED, MD, and anapole modes. The purple dashed line is the sum of the ED and MD fitting curves. The black dashed line indicates the A excitonic resonance. **e** The electric field distributions of the corresponding ED, MD, and anapole modes in (**d**). **f, g** Zoomed-in extinction spectra from box I in (**b**). The extinction spectra (H1-H7) in (**g**) are shifted vertically for clarity. The purple diamonds and yellow stars indicate the resonant energies of UP and LP. Purple and yellow lines in (**f**) are the fitting curves (fitted by Eq.(1)) for UP and LP. The dark blue dashed and dotted lines in (**f**) indicate the dispersions of anapole mode and A exciton separately. Dashed-dotted line in (**g**) indicates the A excitonic resonance. (**h, i**) Similar to (**f, g**) but for the zoomed-in region from box II in (**c**).

up strategy introduced in this work provides additional design flexibility for photonic devices, rendering it a promising way to realize large-area metasurfaces[35,53] with vdW TMDC materials.

## Methods

### Dielectric constants of the MoS₂ sample

Following the previous work[11,54], we combined the multi-Lorentz oscillator model and transfer matrix method to fit the transmission (extinction) spectrum we obtained in the experiment (Supplementary Fig. 16). The excitonic resonant frequencies and oscillator strengths for A, B, C excitons are adjusted by comparing the numerical and experimental results, to obtain the best fitting. Then the fitting parameters were substituted in the Lorentz model to obtain the dielectric function of bulk MoS₂ in this work.

### Material properties characterizations

The cross-section (*yz*-plane) of MoS₂ structure was obtained by etching the 1D MoS₂ grating (A2) via focused ion beam (FIB) technique

(Thermoscientific Scios 2). The morphology of MoS₂ was characterized using TEM/STEM. The Raman spectroscopies are performed in Horiba-Jobin Yvon LabRAM HR Evolution System equipped with 532 nm green laser, where an objective lens of 50X magnification was applied with a 5-second accumulation time.

### Optical characterizations

The angle-resolved extinction measurements for the samples were carried out with a home-built Fourier imaging setup (Supplementary Fig. 2), which consists of an inverted microscope (Nikon Ti2-U) equipped with a couple of objectives with the same optical parameters (Plan Fluor ELWD 60x, NA = 0.7, Nikon)[25]. The quasi-collimated incident light is focused onto the sample through the former objective (OL1) and the transmitted light from the sample passes through the latter objective (OL2). A Fourier lens (FL) at the side port of the microscope records and sends the back focal plane image (i.e. Fourier image) of the latter objective to infinity. Another tube lens (TL) focuses the Fourier image onto the slit of the imaging spectrometer (Shemrock

500i) coupled with an electron-multiplying charge-coupled device camera (EMCCD, iXon Ultra 888). A linear polarizer between the FL and TL defines the polarization of the beam.

### FDTD simulations

A finite-difference time-domain (FDTD) solver(Lumerical Solutions, Inc.) was used in simulation analyses. The refractive index of $MoS_2$ (Fig. 2e in the main text) is inserted in the material lab in the software. The simulations for 1D $MoS_2$ grating were performed by the 2D FDTD simulation region. The illumination consists of a broadband (400–1000 nm) plane-wave beam, which was incident normally to the substrate. Power transmission monitors were placed at the back side (opposite side to the incident source) of the substrate to detect the transmission spectra. The simulations for 2D $MoS_2$ disk arrays were performed under the similar condition as 2D case but a 3D FDTD simulation region was applied in the simulation. For the angle-resolved spectra (Fig. 4b, e), we varied the incident angle from 0 to 40 degrees with a step of 2 degrees.

## Data availability

The experimental and simulated spectra, near field distributions, Raman spectra, SEM images, STEM data, and HRTEM data that support the findings of this work are available at https://doi.org/10.6084/m9.figshare.20494848. Additional data are provided in the Supplementary Information and are available from the corresponding author upon request.

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

## Acknowledgements

The work is in part supported by Research Grants Council of Hong Kong, particularly, via Grant Nos. AoE/P-701/20, 14203018, N_CUHK438/18, and CUHK Group Research Scheme, and CUHK Postdoctoral Fellowship by Innovation and Technology Commission, Hong Kong SAR Government and partially supported by the National Natural Science Foundation of China (62104165), the Natural Science Foundation of Jiangsu Province (BK20200859, BK20210713), Gusu Youth Leading Talent ZXL2021452, and the Priority Academic Program Development (PAPD) of Jiangsu Higher Education Institutions. K.C thanks the support by National Natural Science Foundation of Guangdong for Distinguished Young Scholars (Grant No. 2018B030306043); Pearl River Talent Plan (Grant No. 2019QN01C109); The Fundamental Research Funds for the Central Universities, Sun Yat-sen University (Grant No. 22qntd0503); State Key Laboratory of Optoelectronic Materials and Technology Independent subject (Grant No. OEMT-2022-ZRC-06).

## Author contributions

Z.F.C., J.B.X., F.H.S. and S.W. conceived the concept. F.H.S., Z.F.C. and Y.Q.Z. designed, fabricated and characterized the samples. S.W. and Z.H.Z performed the optical measurements. F.H.S. made the data analyses and FDTD simulations. J.W.M., K.C. and H.J.C. contributed to the data analyses. All the authors contributed to the paper writing.

## Competing interests

The authors declare no competing interests.
