## [Peer Review File · Nature Communications]

Transition metal dichalcogenide metaphotonic and self-coupled polaritonic platform grown by chemical vapor depositionREVIEWER COMMENTS

Reviewer #1 (Remarks to the Author):

F. Shen et al. reported on the fabrication of MoS₂ nanostructures utilising a bottom-up CVD technique. They did two experiments with these structures: 1) dispersion plot and determination of the coupling strength between the exciton and the optical mode. 2) Measuring the Mie mode. However, the work's originality requires further justification. The MoS₂ metasurface and mie resonator have been reported previously: for instance, the MoS₂ metasurface is illustrated in [M. Nauman et al., Nature Communications 12, 5597 (2021)]. and mie resonance has been demonstrated with the use of similar TMDCs, WS₂ [R. Verre et al., Nature Nanotechnology 14, 679-683 (2019)]. As a result, it seems that the uniqueness is the bottom-up method. In that case, this manuscript requires additional clarity regarding how their study differs from that of R. Ma et al. [Advanced Materials 7, 1902093 (2020)]. R. Ma's work also employs a bottom-up strategy, i.e., Mo sulfidation to generate MoS₂. The authors must supply the uniqueness. Therefore, I suggest the major revision of this paper.

Along with the critical comments above, below is a list of major and minor comments:
<Major>

- Scale bar mismatch: The maximum and minimum values for MoS₂ in Fig 1f are -90.0 nm and 90.0 nm, respectively. However, the line plot (Fig. 1g) indicates a range of +/- 60 nm.
- The paper assumes that after sulfidation, the whole structure is converted to MoS₂. However, the Raman measurement is insufficient evidence. For instance, if the MoS₂ is created just on the outer shell of the Mo disc, it will still exhibit two Raman peaks from the top and bottom.
- Are the authors assuming that the MoS₂ is crystalline? Or is it amorphous? And do they have evidence to support their claim?
- Detailed explanation of the experimental setup is lacking. For instance, how were Fig 2b, Fig 3a, and Fig 4g measured?
- The research claims that the clear anti-crossing behaviour between the lower and middle polariton is visible. However, in Fig. 3a, the middle polarity, represented by blue curves, is invisible. With the current colour scheme, just guidelines are visible.
- According to the report, they saw Rabi separating from line stripe pattern samples. This is unexpected, as they note in the manuscript because strong light-matter coupling favours small resonator sizes. Can the authors cite any other paper in which Rabi splitting was reported using SLR mode?
- Which simulation approach was applied to obtain Fig 4d? How were exciton peaks included in the simulation?
- How did the authors plot the guidelines depicted in Figure 4f? Ip and mp are invisible without those lines.

<Minor>

- The manuscript contains many small errors. Authors must thoroughly proofread the paper. For example, a subscript is frequently missing (TiO₂ -> TiO2)
- Wrong figure index: Figure 2a-> Figure 2b. (line 178)
- Consider rewriting this sentence: "To avoid mess other fitting curves are not shown" (line 207)
- H1-H7 are not visible in Fig 4a.

Reviewer #2 (Remarks to the Author):

In this work, Shen et al. demonstrate a novel approach to bottom-up synthesis of TMDC metasurfaces by using CVD. The approach is interesting, new, and should open exciting possibilities for high-index TMDCs "metaphotonics". The authors demonstrate experimentally novel 1D and 2D arrays of TMDC nanostructures, ribbons and disks, respectively. They also observe self-coupling between optical modes of the system

appearing due to high-index and excitons (both A- and B-) in the very same material. The authors report high-index, up to 4.7, and negligible loss ($<10^{-2}$) in the near-infrared. Experimental data is supported by numerical calculations.

Overall, I find the work of high interest and significant enough novelty and potential impact to grant publication in Nature Communications. However, I think the manuscript would improve if the authors could broaden the introduction section. For example, several previous works have already discussed the metasurface and low-loss applications of TMDCs. Just a fast scan over the web gives several relevant publications, which were not cited by the authors, for example:

1) All-TMD nanophotonics: <https://doi.org/10.1021/acsp Photonics.0c01964>
ACS Photonics 2021, 8, 3, 721–730

2) TMD-metasurfaces with atomic precision:
<https://doi.org/10.1038/s41467-020-18428-2>
Nature Communications volume 11, Article number: 4604 (2020)

Including these and other relevant publications would put this work into a better context.

Reviewer #3 (Remarks to the Author):

The manuscript by Shen et. al. reported realization of large scale MoS₂ metaphotonics structures, fabricated from transitional metal nanopattern followed by CVD sulfidation process to convert the Mo pattern to MoS₂ pattern. Strong coupling between the direct band-gap exciton modes and metaphotonic modes are reported. The fabrication method appears to be novel and the optical quality of the fabricated structure is sufficient to realize strong coupling. It will add to the growing toolbox of engineering exciton-photon interactions of TMDCs.

A few issues that need to be addressed are as follows.

First of all, the authors have neglected a large body of closely related work approaching the subject as “photonic crystals” instead of “metaphotonics”. Although there are important distinctions between these two in some circumstances, not quite in this work. The analysis and understanding described in this work could be equally well done in the context of “photonic crystals”, which is much more mature and well established. For example, there have been a lot of work integrating monolayer TMDCs with dielectric photonic crystal structures with exciton-photon strong coupling (Zhang et al Nature Communications 9, 713 (2018), Liu et al Science 370, 600 (2020) etc.), as well as using the TMDC itself as the photonic crystal medium (Zhang et al Nature Nanotechnology 14, 844 (2019), Zhang et al Advanced Optical Materials 8, 1901988 (2020)). In comparison, the main novelty of the current work is the new fabrication method to realize large scale structures.

It is worth pointing out that monolayer TMDCs placed at the surface of dielectric photonic crystals can couple nearly as strongly as placed at the cavity field maximum. If the coupling strength is found to be larger in this work, it is likely mainly because of a thick TMDCs layer is used instead of a monolayer.

Although it is an interesting approach to use the TMDC itself to construct the photonic structure, it is limited to the direct excitons at a higher energy than the (indirect) band gap. Using it as emission device or to study phase transitions, for example, are largely impractical. The authors should clarify this.

Another concern is the stability of such nanopatterned TMDCs. How does the open edges of the nanostructure impact the stability and optical quality of the system?

On a more technical note, how do the A and B exciton energies and A and B exciton

oscillator strengths, measured or used in the simulations in this work, compare with values reported in the literature?

I found Figure 3 confusing. Why it is spatial dispersion with axis labeled as k ? The measured and simulated data in a and b don't look like what's in d. It is not obvious how one can tell there is strong coupling from a and b. One can always assume there is strong coupling and calculate the dispersion correspondingly and compare with the measurement to check consistency. To go beyond an assumption, there needs to be a clear, distinct signature of the measured dispersion compared to possible dispersions without strong coupling. Such a comparison can be seen in Fig3d. But I have trouble relating Fig 3d to the measured results showing in Fig. 3a.

Similarly, Fig 4 b-c and f-g seem to only compare "what if it is strong coupling", without critically answering "is it possible it is NOT strong coupling".

Response letter for the manuscript entitled
“Transition Metal Dichalcogenide Metaphotonics and Self-coupled
Polariton Platform Realized by CVD Bottom-up Method”

*Blue letters are used to as a guidance for the modification we have made in the revised manuscript and supporting information.

Response to Reviewer #1

We thank the reviewer for the valuable suggestions about our manuscript and appreciate all the constructive criticism. In what follows, we will provide detailed responses to the individual comments.

General comment 1:

The MoS₂ metasurface and Mie resonator have been reported previously: for instance, the MoS₂ metasurface is illustrated in [M. Nauman et al., Nature Communications 12, 5597 (2021)]. and mie resonance has been demonstrated with the use of similar TMDCs, WS₂ [R. Verre et al., Nature Nanotechnology 14, 679-683 (2019)]. As a result, it seems that the uniqueness is the bottom-up method.

Response: We agree with the reviewer that one of the major contributions of our work is applying the CVD bottom-up method to construct TMDCs metaphotonic structures(or metastructures) and self-coupled polariton systems. As an emerging field, previous work majorly utilized the top-down(i.e., etching) method to fabricate the nanostructures from the exfoliated TMDCs flakes, which suffers from the limited lateral sizes and uneven thicknesses[1]. Consequently, the scalability and reproducibility of TMDCs metastructures are hindered by substantial challenges in future applications such as large-scale photonic integrated circuits and metalens arrays, which typically include hundreds of components with a device scale of more than several hundred millimeters or centimeters[2].

Compared to the top-down method, the bottom-up method proposed in this work overcomes the abovementioned challenges. With this bottom-up strategy, the geometric parameters of the structure can be well designed and precisely controlled in the fabrication process, which is key to the performance and the reproducibility of the metaphotonic devices. In addition, there seems no limitation on device size based on this bottom-up method, making the complex photonic device or integrated photonic circuits possible in practice[3]. In this regard, our work, proving some fundamental dielectric resonant modes with high refractive index and low loss TMDCs material, lays a great foundation for the charming concept of all van der waals integrated

metaphotonics, which emerges as an important branch of dielectric photonics.

General comment 2:

In that case, this manuscript requires additional clarity regarding how their study differs from that of R. Ma et al. [Advanced Materials 7, 1902093 (2020)]. R. Ma's work also employs a bottom-up strategy, i.e., Mo sulfidation to generate MoS₂. The authors must supply the uniqueness.

Response: Thanks for the reviewer's question. The major uniqueness of our work is utilizing the CVD bottom-up method to demonstrate the large-scale high-refractive-index metastructures and self-coupled polaritons with bulk TMDCs. To our knowledge, the CVD bottom-up is, for the first time, applied in the fields of dielectric nanophotonics based on bulk TMDCs.

In R.Ma's work[4] they successfully demonstrated the tight contact of few-layer(around 2-4 layers)MoS₂ on the surface of Au disk(Fig. R1), and they claimed that such tight contact could enhance interaction between few-layer MoS₂ and surface plasmon and thus may have the potential applications in improving the performance of photodetectors and photocatalysts. In contrast to it, our work majorly focuses on the fields of dielectric nanophotonics and exhibits uniqueness in the following aspects(we here use **red bold** and **black bold** letters to emphasize the differences between **R.Ma's work** and **our work**):

1. **Bulk TMDCs dielectric metastructures with well-designed geometric parameters**

(In R.Ma's work, they fabricated the **randomly arranged MoS₂@Au-disk** structures with few-layer(**2-4 layers**) MoS₂ covered on the surfaces of the Au disks, as **Fig. R1a** shows. However, our sample are **all-dielectric metastructures** built with **bulk MoS₂** whose geometric parameters are well designed and maintained in the final outputs, as Fig. R1b clearly shows. Meanwhile, in R.Ma's work few-layer MoS₂ only weakly interacts with light which is non-ideal in the dielectric nanophotonics where typically dimension of the dielectric resonator reaches more than 100 nm[5][6], while our bulk TMDCs structure shows much more intense optical responses which help to design the high-performance dielectric photonic devices[7].)

2. **Demonstration of the resonant modes in the high-refractive-index TMDCs metastructures**

(In R.Ma' work, they mainly focused on the **material properties characterization** with Raman and XPS spectra throughout the whole work and claimed potential application in **photodetectors and photocatalysts** with their tightly contacted Au@MoS₂ disk structures. However, in our work, we firstly demonstrate the **high refractive index**(in the visible and infrared region) of MoS₂ which is key to the optical response of the dielectric resonator. Then **optical measurements**, as well as the **simulation**, are utilized to characterize the **strong electric and magnetic modes**

of our structured MoS₂ pattern. In other words, our work majorly focuses the fields of dielectric nanophotonics which is distinct to the main content of R.Ma's work.)

3. **Strong exciton-photon interaction with the concept of self-coupled polaritons**
(In R.Ma's work, they didn't achieve strong light-matter interaction with some dominant signatures such as the Rabi splitting or anti-crossing behavior. However, in our work we achieve the **strong coupling** with the unambiguous **anti-crossing** behavior shown in the manuscript.)

Fig. R1. Schematic figure to show the comparisons of R.Ma et al's work(ref[4]) and our work regarding the structures before and after Mo-to-MoS₂ conversion process. The left panel shows the corresponding SEM figures of sample.

Last but not the least, though in this work we use the CVD bottom-up method to demonstrate the dielectric resonant modes with some proof-of-concept MoS₂ structures, our method can be further extended in multiple fields including the integrated photonic circuits and metalens. Different from the previously used top-down method, our method shows no limitation in the device size and structure design, the large refractive index and low material loss of bulk MoS₂ demonstrated in our work can be further applied to some important photonic components such as waveguide, modulator, and optical resonator, especially for the applications in the near-infrared and telecom frequency range[3].

Revision details:

Related sentences in the revised manuscript to emphasize the uniqueness of our work in contrast to R.Ma's work:

“The all-dielectric metastructures(thickness>100 nm) based on the bulk TMDCs achieved in our work show distinct differences from the Au@MoS₂ structure reported

in the previous work where the few-layer(2-4 layers) MoS₂ is formed at the surface of Au disk after sulfidation. Strong electric and magnetic resonant responses are realized and enhanced in the well-structured high-refractive-index MoS₂ pattern which is key in the fields of metaphotonics.”(page 5, end of 1st paragraph)

Related sentences in the revised manuscript to point out the key role the CVD bottom-up method played in the further development of bulk-TMDCs-based metaphotonics:

“Moreover, our CVD bottom-up method overcomes the obstacles encountered by the previously reported top-down method regarding the design and fabrication of bulk-TMDCs-based metastructures. With this bottom-up method, it is thus capable of realizing the concept of all van der Waals integrated photonics circuits for the applications in near-infrared and telecom frequency range, and the large-area TMDCs metalens array with complex functionality.”(page 15, 1st paragraph)

Major comments 1:

Scale bar mismatch: The maximum and minimum values for MoS₂ in Fig 1f are -90.0 nm and 90.0 nm, respectively. However, the line plot (Fig. 1g) indicates a range of +-60 nm.

Response:

We thank the reviewer to point out the mismatch and we have changed the range of Fig.1d and Fig.1g to ± 90 nm to match the Fig.1c and Fig.1f in the revised manuscript.

Major comments 2:

The paper assumes that after sulfidation, the whole structure is converted to MoS₂. However, the Raman measurement is insufficient evidence. For instance, if the MoS₂ is created just on the outer shell of the Mo disc, it will still exhibit two Raman peaks from the top and bottom.

Response: We agree with the reviewer that the Raman measurements in our work can only characterize the properties of the outer shell of MoS₂ structures. In the revised manuscript, we have supplemented the characterizations of the material and crystal structural properties inside MoS₂ structures. We firstly utilize the focused ion beam(FIB) technique to obtain the cross-section of the MoS₂ grating, as the scanning transmission electron microscopy (STEM) image in Fig. R2(a) shows. Then the energy dispersive X-ray spectroscopy (EDS) is used to acquire the element maps of the cross-section. As Fig. R2(a) exhibits, both Mo and S elements are shown in the boundary and internal regions of the MoS₂ cross-section which unambiguously confirms that the sulfidation process is complete where the whole Mo pattern is converted to MoS₂ pattern.

Revision details:

The STEM image and EDS maps of the cross-section from the MoS₂ grating have been supplemented in Fig. 2a(i.e., as Fig. R2(a) shows) in the revised manuscript to show the element maps of our MoS₂ sample.

Fig. R2. (a) STEM of cross section of 1D MoS₂ grating(left 1); Si, O, Pt, Mo, S element distribution of the cross section and overlay(right 1). (b) HRTEM of three corresponding regions marked by the squares in (a).

Major comments 3:

Are the authors assuming that the MoS₂ is crystalline? Or is it amorphous? And do they have evidence to support their claim?

Response: We thank the reviewer for this meaningful question. Using the same grating cross-section, we apply the high-resolution TEM(HRTEM) to characterize the crystallinity properties of our MoS₂ sample. As Fig. R2(b) shows, our MoS₂ sample exhibit the typical polycrystalline characteristics which is composed of multiple crystallites of varying sizes and orientations. The interval of neighboring MoS₂ layers is around 0.62nm which is consistent with previous results[8]. More interestingly we find that the MoS₂ of around 10 layers prefer to grow horizontally at the surface and the interface between the MoS₂ and substrate(green and magenta boxes) while the internal MoS₂ layers prefer to be vertically aligned to the surface(yellow box). It appears quite interesting to analyze or understand the mechanism behind it but it is far beyond the scope of this work. We will be interested in studying this phenomenon in further investigation.

Revision details:

The HRTEM of the MoS₂ grating cross-section has been supplemented in Fig. 2b(i.e., as Fig. R2b shows) in the revised manuscript to show the crystal structural properties of our MoS₂ sample.

Major comments 4:

Detailed explanation of the experimental setup is lacking. For instance, how were Fig 2b, Fig 3a, and Fig 4g measured?

Response: We thank the reviewer for this question. We have supplemented illustrations of the experimental setup for the optical measurements in the revised manuscript:

“The angle-resolved extinction measurements for the samples were carried out with a home-built Fourier imaging setup, which consists of an inverted microscope (Nikon Ti2-U) equipped with a couple of objectives with the same optical parameters (Plan Fluor ELWD 60x, NA = 0.7, Nikon). The quasi-collimated incident light is focused onto the sample through the former objective (OL1) and the transmitted light from the sample passes through the latter objective (OL2). A Fourier lens (FL) at the side port of the microscope records and sends the back focal plane image (i.e. Fourier image) of the latter objective to infinity. Another tube lens (TL) focuses the Fourier image onto the slit of the imaging spectrometer (Shemrock 500i) coupled with an electron-multiplying charge-coupled device camera (EMCCD, iXon Ultra 888). A linear polarizer between the FL and TL defines the polarization of the beam.”

Fig. R3. Schematic Fourier imaging setup for the measurement of angle-resolved extinction spectra. OL: objective lens, FL: Fourier Lens, DL: delayed lens, TL: tube lens, LP: linear polarizer, PM: parabolic mirror, NA: numerical aperture, BFP: back focal plane.

Revision details:

The methods part including the optical measurements and morphological characterization have been supplemented in the revised manuscript.

Major comments 5:

The research claims that the clear anti-crossing behaviour between the lower and middle polariton is visible. However, in Fig. 3a, the middle polarity, represented by blue curves, is invisible. With the current colour scheme, just guidelines are visible.

Response: We thank referee for pointing out the issues in exhibiting the anti-crossing behaviors of the polaritons in Fig. 4a(corresponding to Fig. 3a in original version). We claim that the anti-crossing behavior of the polaritons do exist in our self-coupled system while some modifications need to be made to clearly show it:

1. Eliminating the interference of B exciton

We notice that the contribution of B exciton to the total coupling process is negligible(comparing to A exciton). Previous works on the coupling between exciton from MoS₂ and cavity modes(ranging from the DBR Fabry-Perot cavity, to the plasmonic cavity or the silicon mie resonator[8][9][10]) focus their discussion on the two-coupled model(i.e., A exciton coupled with cavity mode). In our previous analyses, we also notice that B exciton only shows the minor influence on the total polariton dispersion. As a result, without loss of generality, in the revised manuscript we will considered only two polariton branches(upper polariton or UP; lower polariton or LP) arising from the coupling of A exciton to M-SLR or Mie modes, avoiding the interference from B exciton.

2. Change the colormap

The colormap is modified in Fig. 4a and Fig. 4b(as shown in the Fig. R4(a) and Fig. R4(b)) in the revised manuscript(corresponding to Fig. 3a and Fig. 3b in the original version) to highlight the dispersion of lower and higher polaritons. Moreover, a zoomed-in region for Fig. 4a and Fig. 4b(as Fig. R4(c) shows) is provided to show the detailed anti-crossing behavior of lower and upper polaritons. The fitting procedures are provided in the SI part 2.1.

Fig. R4. (a) New colormap for Fig 4(a) in the revised manuscript. (b) New colormap for Fig 4(b) in the revised manuscript. (c) Zoom-in region of (1) and (2) marked with the white boxes in (a).

Last but not the least, to clearly confirm that our theoretical fittings calculated by the theoretical model(red and yellow curves) could well describe the polariton dispersions in the MoS₂ grating structure, we artificially reduce the linewidth of A

exciton from originally 120meV to 40meV in the simulation. As Fig. R5(a) and R5(b) compares, they both exhibit the similar polariton behaviors while the separation of different polariton branches in Fig. R5(b) becomes more distinct with the reduced linewidth. The polariton behavior is majorly dependent on the oscillator strength of exciton[11], and thus the fitting curves in Fig. R5(a) should also apply to the polariton behaviours in Fig. R5(b), which is exactly the case we find in the final simulation results in Fig. R5.

Fig. R5. The FDTD simulated spatial dispersion of A1-A3 with linewidth of A exciton (a) $\gamma = 120\text{meV}$ and (b) $\gamma = 40\text{meV}$. Other parameters are kept the same in the FDTD simulation.

Revision details:

In revised manuscript, we modified color map and exhibited the zoomed-in region to clearly show the anti-crossing behavior in 1D MoS₂ grating. The modified figures are shown in Fig. 4a-c (i.e., as Fig. R5 shows) in the revised manuscript.

Major comments 6:

According to the report, they saw Rabi separating from line stripe pattern samples. This is unexpected, as they note in the manuscript because strong light-matter coupling favours small resonator sizes. Can the authors cite any other paper in which Rabi splitting was reported using SLR mode?

Response: Thanks for the reviewer's question. In revised manuscript, we have cited some representative papers who achieved the strong coupling between exciton from TMDCs and collective SLR mode such as the metallic photonic nanostructures, dielectric photonic crystal and disk array[12][13][14] (corresponding to the ref[24-26] in revised manuscript).

Major comments 7:

Which simulation approach was applied to obtain Fig 4d? How were exciton peaks included in the simulation?

Response: We thank reviewer for these questions.

1. A finite-difference time-domain (FDTD) solver from Lumerical Inc. is used. Multiple *field profile and power monitors* is used to calculate the transmission spectra and near field distribution of the MoS₂ structure. Details for FDTD simulation for this work can be found in SI part 4.
2. To describe the dielectric function of bulk MoS₂ we apply the multiple Lorentz model(similar to the simulation method introduced in Timur et al's work[1]):

$$\epsilon_{MoS_2} = \epsilon_0 + \sum_{i=1}^3 f_i \frac{\omega_i^2}{\omega_i^2 - \omega^2 - j\gamma_i\omega} \quad (R1.)$$

where $i = 1, 2, 3$ represent responses of the A, B and C excitons. Then the dielectric function is substituted into the material properties of structures we setup in the FDTD simulation.

Revision details:

Simulation method has been added in the methods part in the revised manuscript.

Major comments 8:

How did the authors plot the guidelines depicted in Fig. 4f? lp and mp are invisible without those lines.

Response: We thank reviewer for this question. Similar to our reply to the major comment 5, we here simplify our theoretical model by eliminating the contribution from B exciton in the coupling process. As Fig. R6(a,c) shows, after simplifying the fitting model the anti-crossing behaviour becomes more unambiguous and we also show the corresponding extinction spectra in Fig. R6(b,d) for clarity. Fig. R6(a-d) correspond to the Fig. 5f-i in the revised manuscript.

Fig. R6. (a) The experimental anti-crossing behavior due to the coupling of anapole with A exciton as the function of the filling factor Δ . (b) Extinction spectra corresponding to (a). (c) and (d) are the anti-crossing behavior calculated by FDTD simulation.

Revision details:

In revised manuscript, Fig. 5(f-i)(i.e., as Fig. R6 shows) are modified to unambiguously show the anti-crossing behaviour of exciton-polaritons.

Minor comment 1:

The manuscript contains many small errors. Authors must thoroughly proofread the paper. For example, a subscript is frequently missing (TiO₂ -> TiO2)

Response: We thank reviewer for pointing out the typos we made and we have corrected all the typos we found in our manuscript.

Minor comment 2:

Wrong figure index: Fig. 2a-> Fig. 2b. (line 178)

Response: Thank for reviewer's the reminder and we have corrected it in the revised manuscript.

Minor comment 3:

Consider rewriting this sentence: "To avoid mess other fitting curves are not shown" (line 207)

Response: We thank reviewer for pointing out the inappropriate expression and we have replaced the original sentence with "Here only $\kappa=0$ and $\kappa=0.2$ are shown for clarity" in the caption of Fig. 3c in the revised manuscript.

Minor comment 4:

H1-H7 are not visible in Fig 4a.

Response: We thank reviewer for pointing out the figure issues. We modified the Fig. 5a(corresponding to Fig. 4a in original version) in the revised manuscript so as to exhibit H1-H7 more clearly(as Fig. R7 shows).

Fig. R7. Modified Fig. 5a in revised manuscript to clearly show H1-H7.

Reviewer #2

We thank the reviewer for the valuable suggestions about our manuscript and appreciate all the constructive criticism. In what follows, we will provide detailed responses to the individual comments.

General comments:

Overall, I find the work of high interest and significant enough novelty and potential impact to grant publication in Nature Communications. However, I think the manuscript would improve if the authors could broaden the introduction section. For example, several previous works have already discussed the metasurface and low-loss applications of TMDCs. Just a fast scan over the web gives several relevant publications, which were not cited by the authors, for example:

1) All-TMD nanophotonics: <https://doi.org/10.1021/acsp Photonics.0c01964>
ACS Photonics 2021, 8, 3, 721–730

2) TMD-metasurfaces with atomic precision:
<https://doi.org/10.1038/s41467-020-18428-2>
Nature Communications volume 11, Article number: 4604 (2020)

Response: We thank the referee for a high evaluation of our work and constructive suggestions. Following the suggestions, we have added more discussion about metasurface based on the TMDCs as well as the comparison of our method with previous works in the revised introduction session and cited the relevant publications[3][15](corresponding to ref[32][22] in the revised manuscript).

Reviewer #3

We thank the reviewer for the valuable suggestions about our manuscript and appreciate all the constructive criticism. In what follows, we will provide detailed responses to the individual comments.

General comments:

The fabrication method appears to be novel and the optical quality of the fabricated structure is sufficient to realize strong coupling. It will add to the growing toolbox of engineering exciton-photon interactions of TMDCs.

Response: We thank referee for the positive evaluation of the quality and impact of our work on the field of light-matter interaction based on the TMDCs.

Major comment 1:

First of all, the authors have neglected a large body of closely related work

approaching the subject as “photonic crystals” instead of “metaphotonics”. Although there are important distinctions between these two in some circumstances, not quite in this work. The analysis and understanding described in this work could be equally well done in the context of “photonic crystals”, which is much more mature and well established. For example, there have been a lot of work integrating monolayer TMDCs with dielectric photonic crystal structures with exciton-photon strong coupling (Zhang et al Nature Communications 9, 713 (2018), Liu et al Science 370, 600 (2020) etc.), as well as using the TMDC itself as the photonic crystal medium (Zhang et al Nature Nanotechnology 14, 844 (2019), Zhang et al Advanced Optical Materials 8, 1901988 (2020)). In comparison, the main novelty of the current work is the new fabrication method to realize large scale structures.

Response: We totally agree with the reviewer that the concept of photonic crystals(or PCs) also applies to our work where the MoS₂ nanoresonators(i.e., grating bar and disk) are periodically arranged.

Most work related to PCs focus more attention on discussing the photonic band properties(including the photonic band gap) in the Brillouin zone(for instances, in ref[16][17][18]), and we do analyze the dispersion properties of 1D MoS₂ grating to show the anti-crossing behavior of polaritons. However, in our work we majorly discuss the optical resonant modes(such as magnetic-type surface lattice modes or the Mie modes) along with their interaction to the MoS₂ exciton. In this regard, we think the concept of metaphotonics could be more suitable in our work to emphasize the strong magnetic and electric response of our TMDCs nanostructures, which is absent in the unstructured bulk TMDCs. We also notice that in some recent published work the concepts of metaphotonics or metasurface are adopted, ranging from the 1D perovskite grating[19] to 2D dielectric disk array[20][21], whose structures are similar to us.

Revision details:

In the revised manuscript, we have broadened our discussion with recent works related to TMDCs-PC hybrid system and cited corresponding papers including ones the reviewer mentioned. We also emphasize the advances of our CVD bottom-up method in the field of metaphotonics. Related sentences in revised manuscript are listed below:

“Recent studies have achieved strong light-matter with monolayer TMDCs coupled to the designed photonic crystal(PC) such as 1D grating, 2D disk array and nontrivial hexagonal dielectric structure, observing the spatial dispersion behaviour of PC-exciton-polaritons and the helical nature of the topological polaritons.”(page 2, 2nd paragraph)

“The device size of our TMDCs metastructures can reach $0.38 \times 0.38 \text{ mm}^2$ in scale(no theoretical limitation in device size) and the geometric shapes of constitute components are greatly maintained after sulfidation.”(page 3, 2nd paragraph)

Major comment 2:

It is worth pointing out that monolayer TMDCs placed at the surface of dielectric photonic crystals can couple nearly as strongly as placed at the cavity field maximum. If the coupling strength is found to be larger in this work, it is likely mainly because of a thick TMDCs layer is used instead of a monolayer.

Response: We agree with the reviewer that the large coupling strength demonstrated in our work is majorly ascribed to the usage of thick TMDCs. As reviewer mentioned, increasing the thickness of the TMDCs would greatly enhance the coupling between exciton and photonic resonator as more excitons would be coupled to the system[22]. The bulk TMDCs used in our system would strongly interact with the resonant mode of the TMDCs metastructure itself, showing the strong light-matter interaction.

Major comment 3:

Although it is an interesting approach to use the TMDC itself to construct the photonic structure, it is limited to the direct excitons at a higher energy than the (indirect) band gap. Using it as emission device or to study phase transitions, for example, are largely impractical. The authors should clarify this.

Response: We agree with the reviewer that bulk TMDCs such as WS₂ and MoS₂ shows poor luminescence properties due to the indirect band gap which greatly hinder the future applications for the emission devices. However, our method can be further extended to other TMDCs systems. For instance, the bulk InSe was reported to show the strong photoluminescence at near infrared range[23] which indicates the possibilities to build the emission metadevices with the similar method proposed in this work.

Revision details:

We have added the discussion of the emission properties of our MoS₂ pattern and extension of method to other TMDCs system in the end part of the revised manuscript. Some related sentence is listed below:

“Some bulk materials of direct bandgap such as InSe could be ready to be used for design and fabrication of emission devices.” (page 14, 1st paragraph)

Major comment 4:

Another concern is the stability of such nanopatterned TMDCs. How does the open edges of the nanostructure impact the stability and optical quality of the system?

Response: We thank the reviewer for this meaningful question. We found that the metastructures built with bulk TMDCs are quite stable and open edges would have negligible influence on the optical properties of the system.

As Fig. R8(a) shows, after more than 6 months(stored in ambient environment) the corresponding peaks in Raman spectrum(blue) are almost unchanged, compared to the

original spectrum(red). We find that linewidths of the Raman peaks are slightly broadened which may be due to the surface contamination or the oxidation of the open edges. Moreover, the horizontal-aligned MoS₂ layer at the surface(see HRTEM images in Fig. 2b in the revised manuscript or Fig. R2b) can effectively reduce the chemically reactive edge sites(dangling bonds are greatly reduced) and contribute to the improvement of the stability.

Then we compare the spatial dispersions of original MoS₂ gratings sample and the

same sample after 6 months storage. As Fig. R8b, c shows, the resonant frequency of M-SLR mode and anti-crossing behavior are almost intact after 6 months, indicating the negligible influence of the surface contamination or the oxidation on the optical properties of the MoS₂ structures.

Fig. R8. (a) Original(red) Raman spectra of MoS₂ nanostructure and the spectra after more than 6 months(blue). (b) Spatial dispersion of A1-A3 for original measurements(i.e., Fig. 4a in the revised manuscript) and (c) measurement results after more than 6 months.

Revision details:

We have added the stabilities properties of MoS₂ sample in the revised supporting information(Fig. S18).

Major comment 5:

On a more technical note, how do the A and B exciton energies and A and B exciton oscillator strengths, measured or used in the simulations in this work, compare with values reported in the literature?

Response: Following the previous work[1][24], the dielectric function of the bulk MoS₂ can be described by the multiple Lorentz model(which is inherently match the Kramers-Kronig relation):

$$\varepsilon_{MoS_2} = \varepsilon_0 + \sum_{i=1}^3 f_i \frac{\omega_i^2}{\omega_i^2 - \omega^2 - j\gamma_i\omega} \quad (R2.)$$

where $i = 1, 2, 3$ represent responses of the A, B and C excitons.

The initial values of resonant frequencies as well as the oscillator strengths of A, B, C excitons can be obtained via the ref[24](Fig. 5c and Fig. 5g). However, we noticed that some adjustments need to be done to obtain more accurate permittivity for our bulk MoS₂ samples. To be specific, we measured the transmission spectrum of the MoS₂ film(after sulfidation). Then the transfer matrix method(TMM) is applied to fit the transmission spectrum of the MoS₂ film where the parameters were varied to match the experimental transmission results and the values giving the best fitting were exploited in our work. At last, the resonance frequencies($\omega_1, \omega_2, \omega_3$) as well as the corresponding oscillator strengths(f_1, f_2, f_3) were substituted in the eq(R2) to calculate the dielectric function of the MoS₂. We have given a detailed description of this part in supporting information(part 3.3) for the revised version.

Revision details:

The detailed description of how to obtain the dielectric function(for simulation) of our MoS₂ sample is provided in the revised supporting information(part 3.3).

Major comment 6:

I found Fig. 3 confusing. Why it is spatial dispersion with axis labeled as k? The measured and simulated data in a and b don't look like what's in d. It is not obvious how one can tell there is strong coupling from a and b. One can always assume there is strong coupling and calculate the dispersion correspondingly and compare with the measurement to check consistency. To go beyond an assumption, there needs to be a clear, distinct signature of the measured dispersion compared to possible dispersions without strong coupling. Such a comparison can be seen in Fig3d. But I have trouble relating Fig 3d to the measured results showing in Fig. 3a.

Response:

1. Thanks for the reviewer's question. The spatial dispersion describes the relation between the resonant frequency of the optical mode and the in-plane component of the incident wave vector $\mathbf{k}(=\frac{2\pi}{\lambda})$. The in-plane component of the incident wave vector is defined as $k_{//} = \sin(\theta) \frac{2\pi}{\lambda}$ where θ is incident angle and λ is the wavelength.
2. We simulate the spatial dispersion of A2 by artificially set the material permittivity, i.e., one without A,B excitons(B exciton is neglected in the following discussion) and the other taking A exciton into consideration(Fig. R9) to compare the uncoupled and coupled cases. Here the contribution of C exciton would be kept for both two cases as the background. Fig. R10 clearly exhibits distinct dispersions for the uncoupled M-SLR(without A exciton, Fig. R10a) and the lower&upper polaritons in the coupled system(with A exciton, Fig. R10b).

Fig. R9. Artificial permittivities((a): real part, (b): imaginary part) to simulate transmission spectra for the uncoupled(no A, B excitons) photonic mode and coupled(with A, B excitons) photonic mode.

Fig. R10. Simulated spatial dispersion using the artificial permittivities in Fig. R9. (a) without A, B excitonic response(uncoupled). (b)with A, B excitonic response(coupled). White dashed-dot line marks the position of A exciton.

Revision details:

1. In the revised manuscript, we have replaced k with $k_{//}$ in the revised manuscript(similar to ref[14]) and we also have added an inset figure in Fig. 1a to schematically illustrate the definition of $k_{//}$ in the revised manuscript.
2. In the revised version, Fig. R9 and Fig. R10 has added in the Fig. S2 in the revised supporting information.

Major comment 7:

Similarly, Fig 4 b-c and f-g seem to only compare “what if it is strong coupling”, without critically answering “is it possible it is NOT strong coupling”.

Response: Similarly, Fig. R11 shows the simulated uncoupled anapole mode(Fig. R11a) and lower&upper polaritons(Fig. R11b) due to the coupling using the permittivity without A exciton(Fig. R11a) and with A exciton(Fig. R11b). In the revised version, we have added in Fig. S3(i.e., as Fig. R11 shows) in the supporting information.

Fig. R11. (a) Simulated extinction spectra for disk array without A,B excitonic response(a) and with A,B excitonic response(b). White dashed line in (b) marks the position of A exciton.

References

- [1] R. Verre, D. G. Baranov, B. Munkhbat, J. Cuadra, M. Käll, and T. Shegai, “Transition metal dichalcogenide nanodisks as high-index dielectric Mie nanoresonators,” *Nat. Nanotechnol.*, vol. 14, no. July, pp. 679–684, 2019.
- [2] S. Uenoyama and R. Ota, “ 40×40 Metalens Array for Improved Silicon Photomultiplier Performance,” *ACS Photonics*, vol. 8, no. 6, pp. 1548–1555, 2021.
- [3] H. Ling, R. Li, and A. R. Davoyan, “All van der Waals Integrated Nanophotonics with Bulk Transition Metal Dichalcogenides,” *ACS Photonics*, 2021.
- [4] R. Ma *et al.*, “Direct Integration of Few-Layer MoS₂ at Plasmonic Au Nanostructure by Substrate-Diffusion Delivered Mo,” *Adv. Mater. Interfaces*, vol. 7, no. 8, pp. 1–10, 2020.
- [5] M. V. Rybin *et al.*, “High- Q Supercavity Modes in Subwavelength Dielectric Resonators,” *Phys. Rev. Lett.*, vol. 119, no. 24, pp. 1–5, 2017.
- [6] A. E. Miroshnichenko *et al.*, “Nonradiating anapole modes in dielectric nanoparticles,” *Nat. Commun.*, vol. 6, pp. 1–8, 2015.
- [7] A. I. Kuznetsov, A. E. Miroshnichenko, M. L. Brongersma, Y. S. Kivshar, and B. Luk’yanchuk, “Optically resonant dielectric nanostructures,” *Science (80-.)*, vol. 354, no. 6314, 2016.
- [8] T. Hinamoto *et al.*, “Resonance Couplings in Si @ MoS₂ Core – Shell Architectures,” vol. 2200413, pp. 1–9, 2022.
- [9] X. Liu *et al.*, “Strong light-matter coupling in two-dimensional atomic crystals,” *Nat. Photonics*, vol. 9, no. 1, pp. 30–34, 2014.
- [10] L. Yang *et al.*, “Strong Light–Matter Interactions between Gap Plasmons and Two-Dimensional Excitons under Ambient Conditions in a Deterministic Way,” *Nano Lett.*, 2022.
- [11] J. Sun *et al.*, “Light-Emitting Plexciton: Exploiting Plasmon-Exciton Interaction in the Intermediate Coupling Regime,” *ACS Nano*, vol. 12, pp. 10393–10402, 2018.

- [12] S. Wang *et al.*, “Coherent coupling of WS₂ monolayers with metallic photonic nanostructures at room temperature,” *Nano Lett.*, vol. 16, no. 7, pp. 4368–4374, 2016.
- [13] L. Zhang, R. Gogna, W. Burg, E. Tutuc, and H. Deng, “Photonic-crystal exciton-polaritons in monolayer semiconductors,” *Nat. Commun.*, vol. 9, no. 1, pp. 1–8, 2018.
- [14] S. Wang *et al.*, “Collective Mie Exciton-Polaritons in an Atomically Thin Semiconductor,” *J. Phys. Chem. C*, vol. 124, no. 35, pp. 19196–19203, 2020.
- [15] B. Munkhbat, A. B. Yankovich, D. G. Baranov, R. Verre, E. Olsson, and T. O. Shegai, “Transition metal dichalcogenide metamaterials with atomic precision,” *Nat. Commun.*, vol. 11, no. 1, pp. 1–8, 2020.
- [16] K. Dong *et al.*, “Flat Bands in Magic-Angle Bilayer Photonic Crystals at Small Twists,” *Phys. Rev. Lett.*, vol. 126, no. 22, p. 223601, 2021.
- [17] M. Kaek, R. Damari, M. Roth, S. Fleischer, and T. Schwartz, “Strong Coupling in a Self-Coupled Terahertz Photonic Crystal,” *ACS Photonics*, 2021.
- [18] B. Lou and S. Fan, “Tunable Frequency Filter Based on Twisted Bilayer Photonic Crystal Slabs,” *ACS Photonics*, pp. 1–6, 2022.
- [19] Y. Fan *et al.*, “Enhanced Multiphoton Processes in Perovskite Metasurfaces,” *Nano Lett.*, vol. 21, no. 17, pp. 7191–7197, 2021.
- [20] G. W. Castellanos, P. Bai, and J. Gómez Rivas, “Lattice resonances in dielectric metasurfaces,” *J. Appl. Phys.*, vol. 125, no. 21, 2019.
- [21] A. Komar *et al.*, “Dynamic Beam Switching by Liquid Crystal Tunable Dielectric Metasurfaces,” *ACS Photonics*, vol. 5, no. 5, pp. 1742–1748, 2018.
- [22] S. Wang *et al.*, “Limits to Strong Coupling of Excitons in Multilayer WS₂ with Collective Plasmonic Resonances,” *ACS Photonics*, vol. 6, no. 2, pp. 286–293, 2019.
- [23] G. W. Mudd *et al.*, “Tuning the bandgap of exfoliated InSe nanosheets by quantum confinement,” *Adv. Mater.*, vol. 25, no. 40, pp. 5714–5718, 2013.
- [24] Y. Li *et al.*, “Measurement of the optical dielectric function of monolayer transition-metal dichalcogenides: MoS₂, MoSe₂, WS₂, and WSe₂,” *Phys. Rev. B - Condens. Matter Mater. Phys.*, vol. 90, no. 20, pp. 1–6, 2014.

REVIEWER COMMENTS

Reviewer #1 (Remarks to the Author):

The authors answered comprehensively to all of my major and minor comments; therefore, I recommend that this manuscript be accepted for publication.

Reviewer #3 (Remarks to the Author):

The authors' rebuttal addressed some of the concerns, such as additions of references and data on stability of the device. But some concerns have not been cleared.

The work makes an original contribution in showing metaphotonic structures made of CVD grown MoS₂, and the data and analysis are reasonably clear. But there are numerous overstatements and inaccurate statements in the manuscript. These need to be carefully and thoroughly worked through before the manuscript is appropriate for publication.

Here are some examples:

Abstract:

"The magnetic-type surface lattice resonance(M-SLR) with extremely low material loss..."

I don't think the authors showed extremely low material loss.

"Our CVD bottom-up fabrication method successfully demonstrates the fundamental optical responses with the proof of concept MoS₂ metastructures and paves a new way in the nanophotonics fields such as the integrated photonic circuits and metasurface"

I don't understand what the authors mean by "demonstrats the fundamental optical responses". The "proof of concept MoS₂ metastructures" has been demonstrated before but not with the same method, so for the same reason, I also would not claim what has been demonstrated as "a new way" for "integrated photonic circuits and metasurface".

Introduction:

"However, due to the limited thickness, monolayer or few-layer TMDCs suffer from a weak interaction with light [28][29][30]. In contrast, the bulk TMDCs exhibits great potential to confine and guide light in the subwavelength scale, paving the way to realize the intriguing concept of all van der Waals integrated nanophotonics and metadevices with complex functionalities[31][32]."

Monolayer TMDCs have larger oscillator strength per thickness compared to bulk. Stacking multiple TMDCs can reach stronger oscillator strength than demonstrated here.

More importantly, monolayer TMDCs have a direct bandgap. Bulk TMDCs having an indirect bandgap is a huge limitation for potential photonics applications, as is evident from the choice of active materials in existing photonic devices.

The authors responded to this by adding "Some bulk materials of direct bandgap such as InSe could be ready to be used for design and fabrication of emission devices." But that is not what this paper has demonstrated. On the other hand, if the same way of fabrication is readily applied to arbitrary other materials, then the present work becomes rather trivial, as the main originality is in experimentally implementing the bottom-up method on MoS₂.

"With this top-down method, the device size will be limited by the TMDCs flakes and the corresponding optical performance would suffer from the uneven and uncontrollable flake thickness[35]. Consequently, it impedes sufficient reproducibility of the TMDCs metastructures and hinders the further realization of the complex metadevices such as large-area metalens array[36]. To realize reproduced performance of TMDCs-based photonics devices it is required to fabricate the nanostructures with uniform and controllable geometric properties."

I do not see a clear advantage of the bottom-up method shown in the present manuscript compared to the top-down methods shown in Ref 31-34.

The top-down methods are NOT limited by the TMDCs flakes. Flakes were used in previous works because they offered better optical performance compared to CVD grown TMDCs. The same top-down methods can be used for CVD or other large area TMDCs, and probably easier than for flakes.

As to "uniform and controllable geometric properties", the top-down and bottom-up methods are also similar. Both methods use lithography and etching to define the structures. Did the present work produce better uniformity and controllability compared to previous work? Are there data to support this?

A follow up comment on the "spatial dispersion". Energy dispersion vs. the wavenumber k is not "spatial dispersion". There is no discussion of directional dependence in the manuscript.

There are many other places with ambiguous or inaccurate languages as well as grammar mistakes throughout the manuscript.

Response letter for the manuscript entitled
“Transition Metal Dichalcogenide Metaphotonic and Self-coupled Polaritonic
Platform by a Chemical Vapor Deposition Bottom-up Method”

*Blue letters are used as a guide for the modifications we have made in the revised manuscript and Supplementary information.

Response to Reviewer #3

We thank the reviewer for the valuable suggestions about our manuscript and appreciate all the constructive criticism. In what follows, we will provide detailed responses to the individual comments.

General comment:

The work makes an original contribution in showing metaphotonic structures made of CVD grown MoS₂, and the data and analysis are reasonably clear. But there are numerous overstatements and inaccurate statements in the manuscript. These need to be carefully and thoroughly worked through before the manuscript is appropriate for publication.

Response: We thank the reviewer for the recognition of our work regarding the originality, data, and analysis. We also thank the reviewer for pointing our attention to the inaccuracies and overstatements that appeared in our manuscript, helping us to improve the quality and accuracy of the manuscript. All the inaccurate and misleading statements are carefully corrected and rephrased in the revised manuscript.

Comment 1:

“The magnetic-type surface lattice resonance(M-SLR) with extremely low material loss...” (abstract)

I don't think the authors showed extremely low material loss.

Response: We thank the reviewer for pointing our attention to the overstatement in the abstract. Multilayer MoS₂ in our work didn't show extremely low material loss and the expression “*extremely low material loss*” has been deleted. Indeed, the material loss of multilayer MoS₂ is moderate (estimated as $10^{-2} \sim 10^{-3}$ in the near-infrared range) in comparison to other semiconductors. Table R1 summarizes the extinction coefficients for some traditional semiconductors and multilayer MoS₂.

Material	Range of κ (750nm to 1000nm)	Source
MoS ₂	10^{-2} to 10^{-4}	Haonan Ling et al[1]
Ge	10^{-1}	Nunley et al[2]
Si	10^{-2} to 10^{-4}	Haonan Ling et al[1]
GaAs	10^{-1} to 10^{-5}	Papatrifonos et al[3]

Table R1. Comparisons of extinction coefficients of bulk MoS₂ and some other semiconductors.

Revision details:

Table R1 has been added to the revised Supplementary information (Table S2).

Comment 2:

“Our CVD bottom-up fabrication method successfully demonstrates the fundamental optical responses with the proof of concept MoS₂ metastructures and paves a new way in the nanophotonics fields such as the integrated photonic circuits and metasurface”

I don't understand what the authors mean by “demonstrates the fundamental optical responses”. The “proof of concept MoS₂ metastructures” has been demonstrated before but not with the same method, so for the same reason, I also would not claim what has been demonstrated as “a new way” for “integrated photonic circuits and metasurface”.

Response: We apologize for the ambiguous and kind of confusing expressions in the abstract. We have deleted the ambiguous expressions such as “fundamental optical

responses” and rephrased the statements to indicate the potential contributions or the influence of our CVD bottom-up method to the fields of metaphotonics. Compared to the previously reported TMDCs nanostructures through patterning the exfoliated flakes, the strategy in our work paves the additional route to realize the TMDC nanophotonic structures and enriches the toolbox of engineering exciton-photon interactions of TMDCs.

Revision details:

(1) Revised statement(in the **abstract**) to summarize the content of this work:

“Here, we report a bulk MoS₂ metaphotonic platform realized by a chemical vapor deposition(CVD) bottom-up method, supporting both the pronounced dielectric optical modes and self-coupled polaritons.”

(2) Revised statement(in the **abstract**) to emphasize the potential influence and applications of the CVD bottom-up method:

“We believe that the CVD bottom-up method would open up possibilities to realize realize large-scale TMDC-based photonic devices and enrich the toolbox of engineering exciton-photon interactions of TMDCs.”

*The original statements, whose expressions are ambiguous, have been deleted.

~~“Our CVD bottom-up fabrication method successfully demonstrates the fundamental optical responses with the proof of concept MoS₂ metastructures and paves a new way in the nanophotonics fields such as the integrated photonic circuits and metasurface.”~~

Comment 3:

Introduction:

“However, due to the limited thickness, monolayer or few-layer TMDCs suffer from a weak interaction with light [28][29][30]. In contrast, the bulk TMDCs exhibits great potential to confine and guide light in the subwavelength scale, paving the way to

realize the intriguing concept of all van der Waals integrated nanophotonics and metadevices with complex functionalities[31][32].”

Monolayer TMDCs have larger oscillator strength per thickness compared to bulk. Stacking multiple TMDCs can reach stronger oscillator strength than demonstrated here.

More importantly, monolayer TMDCs have a direct bandgap. Bulk TMDCs having an indirect bandgap is a huge limitation for potential photonics applications, as is evident from the choice of active materials in existing photonic devices.

The authors responded to this by adding “Some bulk materials of direct bandgap such as InSe could be ready to be used for design and fabrication of emission devices.” But that is not what this paper has demonstrated. On the other hand, if the same way of fabrication is readily applied to arbitrary other materials, then the present work becomes rather trivial, as the main originality is in experimentally implementing the bottom-up method on MoS₂.

Response:

(1) We totally agree with the reviewer that monolayer(ML) TMDCs such as WS₂ and MoS₂ have large excitonic oscillator strengths, such that strong couplings have been achieved by coupling ML TMDCs with extra optical resonators such as plasmonic structures and photonic crystals(PCs)[4][5]. The expressions such as “*weak interaction with light*” are misleading in this regard. We have rephrased the statements to avoid ambiguous and misleading expressions. Rephrased statements can be seen in **revision details (1) and (2)** for this comment.

Indeed, through the above statement, what we truly want to illustrate is that ML TMDCs(thickness < 1nm) are not beneficial to building the dielectric resonators **with the ML TMDCs themselves**, to support dielectric resonant modes such as ED, MD, and anapole[6][7]. In the field of dielectric optics, two parameters, namely, the refractive **index(n)** and **dimensions**(such as thickness) of the nanostructures determine the optical scattering properties of the dielectric resonators[8]. The limited thickness of ML TMDCs in the z -direction would greatly impede the ability to manipulate the light

field or the generation of dielectric resonant modes at visible and near-infrared frequencies. As a consequence, multilayer TMDCs (thickness of dozens nm) were adopted in previous work [6][9] and also in our work for the fabrication of TMDCs nanoresonators or metasurfaces.

For example, the MD mode is excited when the effective wavelength of light inside the dielectric resonator becomes comparable to the structural dimension along the propagation direction of the incident wave, i.e., $nh \sim \lambda_{inc}$ [8][10]. Figure R1 shows the electric field distributions of the isolated disk ($n = 4$) with various thicknesses for the incident wavelength of 550 nm. A pronounced signature (circulating electric fields) of MD resonance arises when the thickness h is within the range of 120 nm to 80 nm. However, once the disk is thin down to 1 nm (i.e., $nh \ll \lambda_{inc}$), the incident electric field only shows a negligible change with the existence of the disk (**Figure R1g**).

(2) We also totally agree with the reviewer that the indirect bandgap of the bulk TMDCs is a huge limitation for potential photonics applications. We have added the discussion of this limitation in the revised manuscript, to indicate that the non-luminous TMDCs multilayer will impede further applications such as the emission devices. On the other hand, to construct metastructures supporting dielectric resonant modes such as M-SLR and anapole modes, the thickness of dozens of nm is typically required [6][8]. In this work, we focus on the fabrication method to realize TMDCs-based metastructures with dielectric resonant modes (e.g., M-SLR, ED, and MD) and self-coupled polaritons.

In the last section of the manuscript, we have proposed a prospect that this CVD bottom-up method could be extended to fabricate the metastructures based on the direct bandgap materials such as InSe, to solve the problem of non-luminescence.

Figure R1. (a) The schematic figure of the scattering of an isolated dielectric disk. The disk with a diameter of $2R = 120$ nm is under normal incidence, which is placed in the vacuum environment ($n=1$). (b-g) The calculated electric field distributions at the yz -plane for the wavelength of 550 nm. The thickness of the disk is varied from 120nm(b) to 1nm(g).

Revision details:

(1) Revised statement to illustrate the strong coupling between monolayer TMDC and photonic crystals:

“In particular, monolayer(ML) TMDCs such as MoS_2 or WS_2 are of direct bandgap and exhibit large excitonic oscillator strengths. By coupling ML TMDCs with optical cavities or dielectric photonic crystals(PCs), the half-light half-matter quasiparticles, i.e., exciton-polaritons, will be formed due to the strong exciton-photon interactions[26][27][28].”

(2) Revised statement to illustrate that the monolayer TMDCs weakly interact with light due to less than 1 nm thickness:

“However, the atomic thin nature of ML TMDCs(thickness<1 nm) is nonideal to construct the dielectric resonators by themselves to support optical resonant modes at visible or near-infrared frequencies, due to the limited thickness along the z-direction[29][30]. Compared to monolayer counterparts, albeit lacking strong luminescence due to the indirect bandgap, multilayer TMDCs still exhibit relatively large exciton oscillator strengths and high refractive indices(>4)[30].”

(3) Revised statements to illustrate the limitations of the non-luminous properties of bulk MoS₂ and the prospect:

“In this work, both the 1D and 2D metastructures are built with the non-luminous material, i.e., the multilayer MoS₂, which impedes further applications such as using it as a gain medium in the field of nanolasers. Extending our CVD bottom-up method to some direct bandgap TMDC materials such as InSe is worthy to be explored in the future[37].”

*The original expression is ambiguous and kind of misleading. We have deleted it in the revised manuscript:

~~“Some bulk materials of direct bandgap such as InSe[52] could be ready to be used for the design and fabrication of emitting devices.”~~

(4) Revised statements to emphasize that the CVD bottom-up paves an additional route to realize the TMDCs metastructures and engineer the light-matter interaction of TMDCs:

“Nevertheless, despite the limitation due to the indirect bandgap of bulk MoS₂, the pronounced dielectric resonances and self-coupled polaritons demonstrated in this work indicate the great potential of adopting the CVD bottom-up method for dielectric metaphotonics and engineering light-matter interaction. The bottom-up strategy introduced in this work provides additional design flexibility for photonic devices, rendering it a promising way to realize large-area metasurfaces[35][55] with vdW TMDC materials.”

Comment 4:

“With this top-down method, the device size will be limited by the TMDCs flakes and the corresponding optical performance would suffer from the uneven and uncontrollable flake thickness[35]. Consequently, it impedes sufficient reproducibility of the TMDCs metastructures and hinders the further realization of the complex

metadevices such as large-area metalens array[36]. To realize reproduced performance of TMDCs-based photonics devices it is required to fabricate the nanostructures with uniform and controllable geometric properties.”

I do not see a clear advantage of the bottom-up method shown in the present manuscript compared to the top-down methods shown in Ref 31-34.

The top-down methods are NOT limited by the TMDCs flakes. Flakes were used in previous works because they offered better optical performance compared to CVD grown TMDCs. The same top-down methods can be used for CVD or other large area TMDCs, and probably easier than for flakes.

As to “uniform and controllable geometric properties”, the top-down and bottom-up methods are also similar. Both methods use lithography and etching to define the structures. Did the present work produce better uniformity and controllability compared to previous work? Are there data to support this?

Response:

(1) The major advantage of our bottom-up method is to fabricate scalable nanostructures with controlled thickness, in comparison with the **exfoliated TMDC flakes** used in previous work(**instead of the top-down methods**). We apologize for the misleading statements. We have corrected them in the revised manuscript to show the advantages of our strategy by **compared with the limitations due to exfoliated TMDC flakes**. In the majority of previous work[6][9], nanostructures were constructed by patterning the exfoliated TMDC flakes. The mechanical exfoliation methods are of low productivity and demanding to obtain the large size of TMDC flakes with determined thickness[13]. As a result, the device size of the output nanostructure is limited and the thickness of the structure is hard to control, impeding the sufficient reproducibility of the devices and further large-scale productions. In contrast, the strategy proposed in this work, based on the CVD bottom-up method, shows the **controllable thickness of structures** by controlling the thickness of evaporated Mo film. Moreover, the **large-area** devices(3.8 mm in this work) could be achieved by this bottom-up method.

(2) We also totally agree with the reviewer that the top-down methods are not limited by the TMDC flakes. The limited device size and uncontrolled thickness in previous work are due to the **mechanically exfoliated TMDC flakes**, instead of the top-down methods. We have corrected the misleading expressions and emphasized the limitations are due to the exfoliated flakes instead of the top-down methods.

Although in this work we focus on building the TMDC metastructures with CVD bottom-up method, we believe that top-down methods can also be applied to the large-area multilayer TMDC film (by techniques such as CVD and pulsed laser deposition (PLD)) and are promising to fabricate the TMDC photonic metastructures. To the best of our knowledge, the experimental realization by applying top-down methods to large-area TMDC film (thickness of dozens of nm) has not been reported yet. Some experimental details such as etching conditions for the polycrystalline TMDC films [14] might require further exploration.

(3) Indeed, as the reviewer mentioned, both the bottom-up and top-down methods use lithography to define the nanostructures and they exhibit similar results. We have deleted the original statement (i.e., *“To realize reproduced performance of TMDCs-based photonics devices it is required to fabricate the nanostructures with uniform and controllable geometric properties”*) in the revised manuscript, which turns out to be misleading. In addition, we also have emphasized that the random size and uncontrollable (and uneven) thickness of exfoliated flakes would impede the design flexibility and fabrication of the metastructures [13][15] in the revised manuscript. In contrast, the CVD bottom-up method in this work has realized the scalable MoS₂-based metastructures whose thickness can be controlled by the thickness of evaporated Mo film.

Revision details:

Revised statements to illustrate the advantages of the CVD bottom-up method by compared with the limitations due to the exfoliated flakes in previous work:

“To date, the overwhelming majority of previous reports on TMDC nanostructures are based on the multilayer TMDC flakes by a mechanical exfoliation method[11][24][37]. The exfoliated TMDC flakes are of random sizes and suffer from uncontrollable and uneven thickness[38], thus impeding the sufficient reproducibility and large-scale production of the TMDC metastructures. In contrast, the chemical vapor deposition(CVD) techniques have been proven to be efficient fabrication methods to synthesize large-area TMDCs with controllable thickness[37][39]. It is thus of great promise to adopt the CVD bottom-up method to build scalable photonic devices with TMDC materials, to overcome the limitations due to the exfoliated flakes.”

“Compared to the previous work where the exfoliated WS₂ or MoS₂ flakes were adopted[6][7][19], our bottom-up method achieves the scalable TMDC metastructures with controlled thickness[20].”

*The original statements, which are confusing and misleading, have been deleted:

~~“With this top-down method, the device size will be limited by the TMDCs flakes and the corresponding optical performance would suffer from the uneven and uncontrollable flake thickness[35]. Consequently, it impedes sufficient reproducibility of the TMDCs metastructures and hinders the further realization of the complex metadevices such as large-area metalens array[36]. To realize reproduced performance of TMDCs-based photonics devices it is required to fabricate the nanostructures with uniform and controllable geometric properties.”~~

~~“Distinct from the top-down method based on the exfoliated TMDCs[11][12][37], our method shows the controllable geometric properties of the metastructures. The lateral geometry(as well as the device size) of MoS₂ metastructure can be well defined by EBL technology and the height of MoS₂ structure is determined by the evaporation thickness of Mo pattern[38].”~~

Comment 5:

A follow up comment on the “spatial dispersion”. Energy dispersion vs. the wavenumber k is not “spatial dispersion”. There is no discussion of directional dependence in the manuscript.

Response: We apologize for misusing the term “spatial dispersion”. Instead, we have adopted the term “energy-momentum dispersion” in the main text(which is also adopted in ref[4], [21]), to represent the relation between the energy and the in-plane momentum(i.e., k_x or k_y). In addition, the “polariton dispersions” or “dispersions” that appeared in the revised manuscript also represents the energy-momentum dispersions of the polaritons or optical modes.

The directional dependence of the angle-resolved measurement is shown in Figure R4(a) where the definitions of incident in-plane momentums k_x and k_y are schematically illustrated. Additionally, we define the polarization of incident light with transverse-electric (TE) and transverse-magnetic (TM), depending on whether the electric component is along and across the grating bar. For instance, Figure R4b shows the energy-momentum dispersion of MoS₂ grating(A3) in k_x under TM polarization, where the typical hyperbolic dispersion is shown(compared to the linear dispersion in k_y in Figure 4a of the main text)[4].

Figure R4. (a) A schematic figure to illustrate the in-plane component k_x and k_y of the incident wave vector. (b) The energy-momentum dispersion of MoS₂(grating) in the k_x direction.

Revision details:

Figure R4a has been supplemented in the revised manuscript as Figure 3a. Figure S11 (Figure R4 is included within it) has been added to the Supplementary information to show the dispersions for different propagation directions and polarizations of incident light.

Comment 6:

There are many other places with ambiguous or inaccurate languages as well as grammar mistakes throughout the manuscript.

Response: We apologized for all the inaccurate statements and typos in the manuscript and we have thoroughly proofread the whole manuscript.

References:

- [1] H. Ling, R. Li, and A. R. Davoyan, “All van der Waals Integrated Nanophotonics with Bulk Transition Metal Dichalcogenides,” *ACS Photonics*, 2021.
- [2] T. N. Nunley *et al.*, “Optical constants of germanium and thermally grown germanium dioxide from 0.5 to 6.6eV via a multisample ellipsometry investigation,” *J. Vac. Sci. Technol. B, Nanotechnol. Microelectron. Mater. Process. Meas. Phenom.*, vol. 34, no. 6, p. 061205, 2016.
- [3] K. Papatryfonos *et al.*, “Refractive indices of MBE-grown Al_xGa_(1-x)As ternary alloys in the transparent wavelength region,” *AIP Adv.*, vol. 11, no. 2, 2021.
- [4] L. Zhang, R. Gogna, W. Burg, E. Tutuc, and H. Deng, “Photonic-crystal exciton-polaritons in monolayer semiconductors,” *Nat. Commun.*, vol. 9, no. 1, pp. 1–8, 2018.
- [5] S. Wang *et al.*, “Coherent coupling of WS₂ monolayers with metallic photonic nanostructures at room temperature,” *Nano Lett.*, vol. 16, no. 7, pp. 4368–4374, 2016.
- [6] R. Verre, D. G. Baranov, B. Munkhbat, J. Cuadra, M. Käll, and T. Shegai, “Transition metal dichalcogenide nanodisks as high-index dielectric Mie nanoresonators,” *Nat. Nanotechnol.*, vol. 14, no. July, pp. 679–684, 2019.
- [7] T. D. Green, D. G. Baranov, B. Munkhbat, R. Verre, T. Shegai, and M. Käll, “Optical material anisotropy in high-index transition metal dichalcogenide Mie

- nanoresonators,” *Optica*, vol. 7, no. 6, p. 680, 2020.
- [8] A. I. Kuznetsov, A. E. Miroshnichenko, M. L. Brongersma, Y. S. Kivshar, and B. Luk'yanchuk, “Optically resonant dielectric nanostructures,” *Science* (80-.), vol. 354, no. 6314, 2016.
- [9] M. Nauman *et al.*, “Tunable unidirectional nonlinear emission from transition-metal-dichalcogenide metasurfaces,” *Nat. Commun.*, vol. 12, no. 1, pp. 1–11, 2021.
- [10] I. Staude *et al.*, “Tailoring directional scattering through magnetic and electric resonances in subwavelength silicon nanodisks,” *ACS Nano*, vol. 7, no. 9, pp. 7824–7832, 2013.
- [11] S. Wang *et al.*, “Collective Mie Exciton-Polaritons in an Atomically Thin Semiconductor,” *J. Phys. Chem. C*, vol. 124, no. 35, pp. 19196–19203, 2020.
- [12] W. Liu *et al.*, “Generation of helical topological exciton-polaritons,” *Science* (80-.), vol. 370, no. 6516, pp. 600–604, 2020.
- [13] S. Masubuchi *et al.*, “Autonomous robotic searching and assembly of two-dimensional crystals to build van der Waals superlattices,” *Nat. Commun.*, vol. 9, no. 1, pp. 4–6, 2018.
- [14] M. I. Serna *et al.*, “Large-Area Deposition of MoS₂ by Pulsed Laser Deposition with in Situ Thickness Control,” *ACS Nano*, vol. 10, no. 6, pp. 6054–6061, 2016.
- [15] S. Busschaert, R. Reimann, M. Cavigelli, R. Khelifa, A. Jain, and L. Novotny, “Transition Metal Dichalcogenide Resonators for Second Harmonic Signal Enhancement,” *ACS Photonics*, vol. 7, no. 9, pp. 2482–2488, 2020.
- [16] B. Munkhbat, B. Küçüköz, D. G. Baranov, T. J. Antosiewicz, and T. O. Shegai, “Nanostructured transition metal dichalcogenide multilayers for advanced nanophotonics,” pp. 1–21, 2022.
- [17] W. Huang, L. Gan, H. Li, Y. Ma, and T. Zhai, “Phase-Engineered Growth of Ultrathin InSe Flakes by Chemical Vapor Deposition for High-Efficiency Second Harmonic Generation,” *Chem. - A Eur. J.*, vol. 24, no. 58, pp. 15678–15684, 2018.
- [18] R. Ma *et al.*, “Direct Integration of Few-Layer MoS₂ at Plasmonic Au Nanostructure by Substrate-Diffusion Delivered Mo,” *Adv. Mater. Interfaces*, vol. 7, no. 8, pp. 1–10, 2020.
- [19] X. Zhang *et al.*, “Azimuthally Polarized and Unidirectional Excitonic Emission from Deep Subwavelength Transition Metal Dichalcogenide Annular Heterostructures,” *ACS Photonics*, 2021.
- [20] S. Jang, S. J. Kim, H. J. Koh, D. H. Jang, S. Y. Cho, and H. T. Jung, “Highly Periodic Metal Dichalcogenide Nanostructures with Complex Shapes, High Resolution, and High Aspect Ratios,” *Adv. Funct. Mater.*, vol. 27, no. 46, pp. 1–8, 2017.
- [21] T. Byrnes, N. Y. Kim, and Y. Yamamoto, “Exciton-polariton condensates,” *Nature Physics*, vol. 10, no. 11. Nature Publishing Group, pp. 803–813, 05-Nov-2014.

REVIEWERS' COMMENTS

Reviewer #3 (Remarks to the Author):

The authors have corrected all the misleading statements or over-statements I pointed out and the manuscript is appropriate for publication.